# On Regularizing Rademacher Observation Losses

**Richard Nock**
Data61, The Australian National University & The University of Sydney
richard.nock@data61.csiro.au

## Abstract

It has recently been shown that supervised learning linear classifiers with two of the most popular losses, the logistic and square loss, is equivalent to optimizing an equivalent loss over sufficient statistics about the class: *Rademacher observations* (rados). It has also been shown that learning over rados brings solutions to two prominent problems for which the state of the art of learning from examples can be comparatively inferior and in fact less convenient: (i) protecting and learning from private examples, (ii) learning from distributed datasets without entity resolution. *Bis repetita placent*: the two proofs of equivalence are different and rely on specific properties of the corresponding losses, so whether these can be unified and generalized inevitably comes to mind. This is our *first* contribution: we show how they can be fit into the same theory for the equivalence between example and rado losses. As a *second* contribution, we show that the generalization unveils a surprising new connection to *regularized* learning, and in particular a sufficient condition under which regularizing the loss over examples is equivalent to *regularizing the rados* (*i.e.* the data) in the equivalent rado loss, in such a way that an efficient algorithm for one regularized rado loss may be as efficient when *changing* the regularizer. This is our *third* contribution: we give a *formal* boosting algorithm for the regularized exponential rado-loss which boost with any of the ridge, lasso, SLOPE, $\ell_\infty$, or elastic net regularizer, using the *same* master routine for all. Because the regularized exponential rado-loss is the equivalent of the *regularized logistic loss* over examples we obtain the first efficient proxy to the minimization of the regularized logistic loss over examples using such a wide spectrum of regularizers. Experiments with a readily available code display that regularization significantly improves rado-based learning and compares favourably with example-based learning.

## 1 Introduction

What kind of data should we use to train a supervised learner ? A recent result has shown that minimising the popular logistic loss over examples with linear classifiers (in supervised learning) is equivalent to the minimisation of the exponential loss over *sufficient statistics about the class* known as *Rademacher observations* (rados, [Nock et al., 2015]), for the *same* classifier. In short, we fit a classifier over data that is different from examples, and the same classifier generalizes well to new *observations*. It has been shown that rados offer solutions for two problems for which the state of the art involving examples can be comparatively significantly inferior:

- protection of the examples' privacy from various algebraic, geometric, statistical and computational standpoints, and learning from private data [Nock et al., 2015];
- learning from a large number of distributed datasets *without* having to perform entity resolution between datasets [Patrini et al., 2016].

Quite remarkably, the training time of the algorithms involved can be smaller than it would be on examples, by orders of magnitude [Patrini et al., 2016]. Two key problems remain however: the

accuracy of learning from rados can compete experimentally with that of learning from examples, yet there is a gap to reduce for rados to be not just a good material to learn from in a privacy/distributed setting, but also a serious alternative to learning from examples *at large*, yielding new avenues to supervised learning. Second, theoretically speaking, it is now known that *two* widely popular losses over examples admit an equivalent loss in the rado world: the logistic loss and the square loss [Nock et al., 2015, Patrini et al., 2016]. This inevitably suggests that this property may hold for more losses, yet barely anything displays patterns of generalizability in the existing proofs.

**Our contributions**: in this paper, we provide answers to these two questions, with three main contributions. Our first contribution is to show that this generalization indeed holds: other example losses admit equivalent losses in the rado world, meaning in particular that their minimiser classifier is the *same*, regardless of the dataset of examples. The technique we use exploits a two-player zero sum game representation of convex losses, that has been very useful to analyse boosting algorithms [Schapire, 2003, Telgarsky, 2012], with one key difference: payoffs are non-linear convex, eventually non-differentiable. These also resemble the entropic dual losses [Reid et al., 2015], with the difference that we do not enforce conjugacy over the simplex. The conditions of the game are slightly different for examples and rados. We provide necessary and sufficient conditions for the resulting losses over examples and rados to be equivalent. Informally, equivalence happens iff the convex functions of the games satisfy a symmetry relationship *and* the weights satisfy a linear system of equations. Some popular losses fit in the equivalence [Nair and Hinton, 2010, Gentile and Warmuth, 1998, Nock and Nielsen, 2008, Telgarsky, 2012, Vapnik, 1998, van Rooyen et al., 2015].

Our second contribution came unexpectedly through this equivalence. Regularizing a loss is standard in machine learning [Bach et al., 2011]. We show a sufficient condition for the equivalence under which regularizing the example loss is equivalent to regularizing *the rados* in the equivalent rado loss, *i.e.* making a Minkowski sum of the rado set with a classifier-based set. This property is *independent* of the regularizer, and incidentally happens to hold for *all* our cases of equivalence (*Cf* first contribution). A regularizer added to a loss over examples thus transfers to *data* in the rado world, in essentially the *same* way for all regularizers, and if one can solve the non-trivial computational and optimization problem that poses this data modification for one regularized rado loss, then, basically,

*"A good optimization algorithm for this regularized rado loss may fit to other regularizers as well"*

Our third contribution exemplifies this. We propose an iterative boosting algorithm, $\Omega$-R.ADABOOST, that learns a classifier from rados using the exponential regularized rado loss, with regularization choice belonging to the ridge, lasso, $\ell_\infty$, or the recently coined SLOPE [Bogdan et al., 2015]. Since rado regularization would theoretically require to modify data at each iteration, such schemes are computationally non-trivial. We show that this modification can in fact be bypassed for the exponential rado loss, and the algorithm, $\Omega$-R.ADABOOST, is as fast as ADABOOST. $\Omega$-R.ADABOOST has however a key advantage over ADABOOST that to our knowledge is new in the boosting world: for *any* of these four regularizers, $\Omega$-R.ADABOOST is a boosting algorithm — thus, because of the equivalence between the minimization of the logistic loss over examples and the minimization of the exponential rado loss, $\Omega$-R.ADABOOST is in fact an efficient proxy to boost the *regularized logistic loss over examples* using *whichever* of the four regularizers, and by extension, linear combination of them (*e.g.*, elastic net regularization [Zou and Hastie, 2005]). We are not aware of any regularized logistic loss formal boosting algorithm with such a wide spectrum of regularizers. Extensive experiments validate this property: $\Omega$-R.ADABOOST is all the better vs ADABOOST (unregularized or regularized) as the domain gets larger, and is able to rapidly learn both accurate and sparse classifiers, making it an especially good contender for supervised learning at large on big domains.

The rest of this paper is as follows. Sections §2, 3 and 4 respectively present the equivalence between example and rado losses, its extension to regularized learning and $\Omega$-R.ADABOOST. §5 and 6 respectively present experiments, and conclude. In order not to laden the paper's body, a Supplementary Material (SM) contains the proofs and additional theoretical and experimental results.

## 2    Games and equivalent example/rado losses

To avoid notational load, we briefly present our learning setting to point the key quantity in our formulation of the general two players game. Let $[m] \doteq \{1, 2, ..., m\}$ and $\Sigma_m \doteq \{-1, 1\}^m$, for $m > 0$. The classical (batch) supervised learner is example-based: it is given a set of examples $S = \{(\boldsymbol{x}_i, y_i), i \in [m]\}$ where $\boldsymbol{x}_i \in \mathbb{R}^d$, $y_i \in \Sigma_1$, $\forall i \in [m]$. It returns a classifier $h : \mathbb{R}^d \to \mathbb{R}$ from

a predefined set $\mathcal{H}$. Let $z_i(h) \doteq yh(\boldsymbol{x}_i)$ and abbreviate $\boldsymbol{z}(h)$ by $\boldsymbol{z}$ for short. The learner fits $h$ to the minimization of a *loss*. Table 1, column $\ell_e$, presents some losses that can be used: we remark that $h$ appears only through $\boldsymbol{z}$, so let us consider in this section that the learner rather fits vector $\boldsymbol{z} \in \mathbb{R}^m$.

We can now define our two players game setting. Let $\varphi_e : \mathbb{R} \to \mathbb{R}$ and $\varphi_r : \mathbb{R} \to \mathbb{R}$ two convex and lower-semicontinuous *generators*. We define functions $L_e : \mathbb{R}^m \times \mathbb{R}^m \to \mathbb{R}$ and $L_r : \mathbb{R}^{2^m} \times \mathbb{R}^m \to \mathbb{R}$:

$$L_e(\boldsymbol{p}, \boldsymbol{z}) \doteq \sum_{i \in [m]} p_i z_i + \mu_e \sum_{i \in [m]} \varphi_e(p_i) \ , \tag{1}$$

$$L_r(\boldsymbol{q}, \boldsymbol{z}) \doteq \sum_{\mathcal{I} \subseteq [m]} q_{\mathcal{I}} \sum_{i \in \mathcal{I}} z_i + \mu_r \sum_{\mathcal{I} \subseteq [m]} \varphi_r(q_{\mathcal{I}}) \ , \tag{2}$$

where $\mu_e, \mu_r > 0$ do not depend on $\boldsymbol{z}$. For the notation to be meaningful, the coordinates in $\boldsymbol{q}$ are assumed (wlog) to be in bijection with $2^{[m]}$. The dependence of both problems in their respective *generators* is implicit and shall be clear from context. The adversary's goal is to fit

$$\boldsymbol{p}^*(\boldsymbol{z}) \doteq \arg\min_{\boldsymbol{p} \in \mathbb{R}^m} L_e(\boldsymbol{p}, \boldsymbol{z}) \ , \tag{3}$$

$$\boldsymbol{q}^*(\boldsymbol{z}) \doteq \arg\min_{\boldsymbol{q} \in \mathbb{H}^{2^m}} L_r(\boldsymbol{q}, \boldsymbol{z}) \ , \tag{4}$$

with $\mathbb{H}^{2^m} \doteq \{\boldsymbol{q} \in \mathbb{R}^{2^m} : \mathbf{1}^\top \boldsymbol{q} = 1\}$, so as to attain

$$\mathcal{L}_e(\boldsymbol{z}) \doteq L_e(\boldsymbol{p}^*(\boldsymbol{z}), \boldsymbol{z}) \ , \tag{5}$$

$$\mathcal{L}_r(\boldsymbol{z}) \doteq L_r(\boldsymbol{q}^*(\boldsymbol{z}), \boldsymbol{z}) \ , \tag{6}$$

and let $\partial \mathcal{L}_e(\boldsymbol{z})$ and $\partial \mathcal{L}_r(\boldsymbol{z})$ denote their subdifferentials. We view the learner's task as the problem of maximising the corresponding problems in eq. (5) (with examples; this is already sketched above) or (6) (with what we shall call Rademacher observations, or *rados*), or equivalently minimising negative the corresponding function, and then resort to a *loss function*. The question of when these two problems are equivalent from the learner's standpoint motivates the following definition.

**Definition 1** *Two generators $\varphi_e, \varphi_r$ are said **proportionate** iff $\forall m > 0$, there exists $(\mu_e, \mu_r)$ such that*

$$\mathcal{L}_e(\boldsymbol{z}) = \mathcal{L}_r(\boldsymbol{z}) + b \ , \forall \boldsymbol{z} \in \mathbb{R}^m \ . \tag{7}$$

($b$ does not depend on $\boldsymbol{z}$) $\forall m \in \mathbb{N}_*$, let

$$\mathrm{G}_m \doteq \begin{bmatrix} \mathbf{0}_{2^{m-1}}^\top & \mathbf{1}_{2^{m-1}}^\top \\ \mathrm{G}_{m-1} & \mathrm{G}_{m-1} \end{bmatrix} \ (\in \{0, 1\}^{m \times 2^m}) \tag{8}$$

if $m > 1$, and $\mathrm{G}_1 \doteq [0 \ 1]$ otherwise (notation $\boldsymbol{z}_d$ indicates a vector in $\mathbb{R}^d$).

**Theorem 2** *$\varphi_e, \varphi_r$ are proportionate iff the optima $\boldsymbol{p}^*(\boldsymbol{z})$ and $\boldsymbol{q}^*(\boldsymbol{z})$ to eqs (3) and (4) satisfy:*

$$\boldsymbol{p}^*(\boldsymbol{z}) \in \partial \mathcal{L}_r(\boldsymbol{z}) \ , \tag{9}$$

$$\mathrm{G}_m \boldsymbol{q}^*(\boldsymbol{z}) \in \partial \mathcal{L}_e(\boldsymbol{z}) \ . \tag{10}$$

*If $\varphi_e, \varphi_r$ are differentiable and strictly convex, they are proportionate iff $\boldsymbol{p}^*(\boldsymbol{z}) = \mathrm{G}_m \boldsymbol{q}^*(\boldsymbol{z})$.*

We can alleviate the fact that convexity is strict, which results in a set-valued identity for $\varphi_e, \varphi_r$ to be proportionate. This gives a necessary and sufficient condition for two generators to be proportionate. It does not say how to construct one from the other, if possible. We now show that it is indeed possible and prune the search space: if $\varphi_e$ is proportionate to some $\varphi_r$, then it has to be a "symmetrized" version of $\varphi_r$, according to the following definition.

**Definition 3** *Let $\varphi_r$ s.t. $\mathrm{dom} \varphi_r \supseteq (0, 1)$. $\varphi_{s(r)}(z) \doteq \varphi_r(z) + \varphi_r(1 - z)$ is the **symmetrisation** of $\varphi_r$.*

**Lemma 4** *If $\varphi_e$ and $\varphi_r$ are proportionate, then $\varphi_e(z) = (\mu_r/\mu_e) \cdot \varphi_{s(r)}(z) + (b/\mu_e)$ ($b$ is in (7)).*

| # | $\ell_{\mathsf{e}}(\bm{z}, \mu_{\mathsf{e}})$ | $\ell_{\mathsf{r}}(\bm{z}, \mu_{\mathsf{r}})$ | $\varphi_{\mathsf{r}}(z)$ | $\mu_{\mathsf{e}}$ and $\mu_{\mathsf{r}}$ | $a_{\mathsf{e}}$ |
|---|---|---|---|---|---|
| I | $\sum_{i\in[m]}\log\left(1+\exp\left(z_i^{\mathsf{e}}\right)\right)$ | $\sum_{\mathfrak{I}\subseteq[m]}\exp\left(z_{\mathfrak{I}}^{\mathsf{r}}\right)$ | $z\log z - z$ | $\forall \mu_{\mathsf{e}} = \mu_{\mathsf{r}}$ | $\mu_{\mathsf{e}}$ |
| II | $\sum_{i\in[m]}\left(1+z_i^{\mathsf{e}}\right)^2$ | $-\left(\mathbb{E}_{\mathfrak{I}}\left[-z_{\mathfrak{I}}^{\mathsf{r}}\right] - \mu_{\mathsf{r}}\cdot\mathbb{V}_{\mathfrak{I}}\left[-z_{\mathfrak{I}}^{\mathsf{r}}\right]\right)$ | $(1/2)\cdot z^2$ | $\forall \mu_{\mathsf{e}} = \mu_{\mathsf{r}}$ | $\mu_{\mathsf{e}}/4$ |
| III | $\sum_{i\in[m]}\max\left\{0, z_i^{\mathsf{e}}\right\}$ | $\max\left\{0, \max_{\mathfrak{I}\subseteq[m]}\{z_{\mathfrak{I}}^{\mathsf{r}}\}\right\}$ | $\chi_{[0,1]}(z)$ | $\forall \mu_{\mathsf{e}}, \mu_{\mathsf{r}}$ | $\mu_{\mathsf{e}}$ |
| IV | $\sum_i z_i^{\mathsf{e}}$ | $\mathbb{E}_{\mathfrak{I}}\left[z_{\mathfrak{I}}^{\mathsf{r}}\right]$ | $\chi_{\left[\frac{1}{2^m}, \frac{1}{2}\right]}(z)$ | $\forall \mu_{\mathsf{e}}, \mu_{\mathsf{r}}$ | $\mu_{\mathsf{e}}$ |

Table 1: Examples of equivalent example and rado losses. Names of the rado-losses $\ell_{\mathsf{r}}(\bm{z}, \mu_{\mathsf{r}})$ are respectively the Exponential (I), Mean-variance (II), ReLU (III) and Unhinged (IV) rado loss. We use shorthands $z_i^{\mathsf{e}} \doteq -(1/\mu_{\mathsf{e}})\cdot z_i$ and $z_{\mathfrak{I}}^{\mathsf{r}} \doteq -(1/\mu_{\mathsf{r}})\cdot\sum_{i\in\mathfrak{I}} z_i$. Parameter $a_{\mathsf{e}}$ appears in eq. (14). Column "$\mu_{\mathsf{e}}$ and $\mu_{\mathsf{r}}$" gives the constraints for the equivalence to hold. $\mathbb{E}_{\mathfrak{I}}$ and $\mathbb{V}_{\mathfrak{I}}$ are the expectation and variance over uniform sampling in sets $\mathfrak{I}\subseteq[m]$ (see text for details).

To summarize, $\varphi_{\mathsf{e}}$ and $\varphi_{\mathsf{r}}$ are proportionate iff (i) they meet the structural property that $\varphi_{\mathsf{e}}$ is (proportional to) the symmetrized version of $\varphi_{\mathsf{r}}$ (according to Definition 3), and (ii) the optimal solutions $\bm{p}^*(\bm{z})$ and $\bm{q}^*(\bm{z})$ to problems (1) and (2) satisfy the conditions of Theorem 2. Depending on the direction, we have two cases to craft proportionate generators. First, if we have $\varphi_{\mathsf{r}}$, then necessarily $\varphi_{\mathsf{e}} \propto \varphi_{\mathsf{s(r)}}$ so we merely have to check Theorem 2. Second, if we have $\varphi_{\mathsf{e}}$, then it matches Definition 3[1]. In this case, we have to find $\varphi_{\mathsf{r}} = f + g$ where $g(z) = -g(1-z)$ and $\varphi_{\mathsf{e}}(z) = f(z) + f(1-z)$. We now come back to $\mathcal{L}_{\mathsf{e}}(\bm{z})$, $\mathcal{L}_{\mathsf{r}}(\bm{z})$ (Definition 1), and make the connection with example and rado losses. In the next definition, an e-loss $\ell_{\mathsf{e}}(\bm{z})$ is a function defined over the coordinates of $\bm{z}$, and a r-loss $\ell_{\mathsf{r}}(\bm{z})$ is a function defined over the subsets of sums of coordinates. Functions can depend on other parameters as well.

**Definition 5** *Suppose e-loss $\ell_e(\bm{z})$ and r-loss $\ell_r(\bm{z})$ are such that there exist (i) $f_e : \mathbb{R} \to \mathbb{R}$ and $f_r(z) : \mathbb{R} \to \mathbb{R}$ both strictly increasing and such that $\forall \bm{z} \in \mathbb{R}^m$,*

$$-\mathcal{L}_e(\bm{z}) = f_e\left(\ell_e(\bm{z})\right) \;, \tag{11}$$
$$-\mathcal{L}_r(\bm{z}) = f_r\left(\ell_r(\bm{z})\right) \;, \tag{12}$$

*where $\mathcal{L}_e(\bm{z})$ and $\mathcal{L}_r(\bm{z})$ are defined via two proportionate generators $\varphi_e$ and $\varphi_r$ (Definition 1). Then the couple $(\ell_e, \ell_r)$ is called a couple of equivalent example-rado losses.*

Following is the main Theorem of this Section, which summarizes all the cases of equivalence between example and rado losses, and shows that the theory developed on example / rado losses with proportionate generators encompasses the specific proofs and cases already known [Nock et al., 2015, Patrini et al., 2016]. Table 1 also displays generator $\varphi_{\mathsf{r}}$.

**Theorem 6** *In each row of Table 1, $\ell_e(\bm{z}, \mu_e)$ and $\ell_r(\bm{z}, \mu_r)$ are equivalent for $\mu_e$ and $\mu_r$ as indicated.*

The proof (SM, Subsection 2.3) details for each case the proportionate generators $\varphi_{\mathsf{e}}$ and $\varphi_{\mathsf{r}}$.

## 3 Learning with (rado) regularized losses

We now detail further the learning setting. In the preceeding Section, we have definef $z_i(h) \doteq y h(\bm{x}_i)$, which we plug in the losses of Table 1 to obtain the corresponding example and rado losses. Losses simplify conveniently when $\mathcal{H}$ consists of linear classifiers, $h(\bm{x}) \doteq \bm{\theta}^\top \bm{x}$ for some $\bm{\theta} \in \Theta \subseteq \mathbb{R}^d$. In this case, the example loss can be described using edge vectors $\mathcal{S}_{\mathsf{e}} \doteq \{y_i \cdot \bm{x}_i, i = 1, 2, ..., m\}$ since $z_i = \bm{\theta}^\top(y_i \cdot \bm{x}_i)$, and the rado loss can be described using rademacher observations [Nock et al., 2015], since $\sum_{i\in\mathfrak{I}} z_i = \bm{\theta}^\top \pi_{\bm{\sigma}}$ for $\sigma_i = y_i$ iff $i \in \mathfrak{I}$ (and $-y_i$ otherwise) and $\pi_{\bm{\sigma}} \doteq (1/2)\cdot\sum_i(\sigma_i + y_i)\cdot\bm{x}_i$. Let us define $\mathcal{S}_{\mathsf{r}}^* \doteq \{\pi_{\bm{\sigma}}, \bm{\sigma} \in \Sigma_m\}$ the set of all rademacher observations. We rewrite any couple of equivalent example and rado losses as $\ell_{\mathsf{e}}(\mathcal{S}_{\mathsf{e}}, \bm{\theta})$ and $\ell_{\mathsf{r}}(\mathcal{S}_{\mathsf{r}}^*, \bm{\theta})$ respectively[2], omitting parameters $\mu_{\mathsf{e}}$ and $\mu_{\mathsf{r}}$, assumed to be fixed beforehand for the equivalence to hold (see Table 1). Let us regularize the example loss, so that the learner's goal is to minimize

$$\ell_{\mathsf{e}}(\mathcal{S}_{\mathsf{e}}, \bm{\theta}, \Omega) \doteq \ell_{\mathsf{e}}(\mathcal{S}_{\mathsf{e}}, \bm{\theta}) + \Omega(\bm{\theta}) \;, \tag{13}$$

**Algorithm 1** $\Omega$-R.ADABOOST

---

**Input** set of rados $\mathcal{S}_r \doteq \{\boldsymbol{\pi}_1, \boldsymbol{\pi}_2, ..., \boldsymbol{\pi}_n\}$; $T \in \mathbb{N}_*$; parameters $\gamma \in (0, 1)$, $\omega \in \mathbb{R}_+$;
Step 1 : let $\boldsymbol{\theta}_0 \leftarrow \mathbf{0}$, $\boldsymbol{w}_0 \leftarrow (1/n)\mathbf{1}$ ;
Step 2 : **for** $t = 1, 2, ..., T$
     Step 2.1 : call the weak learner: $(\iota(t), r_t) \leftarrow \Omega\text{-WL}(\mathcal{S}_r, \boldsymbol{w}_t, \gamma, \omega, \boldsymbol{\theta}_{t-1})$;
     Step 2.2 : compute update parameters $\alpha_{\iota(t)}$ and $\delta_t$ (here, $\pi_{*k} \doteq \max_j |\pi_{jk}|$):

$$\alpha_{\iota(t)} \leftarrow (1/(2\pi_{*\iota(t)})) \log((1 + r_t)/(1 - r_t)) \quad \text{and} \quad \delta_t \leftarrow \omega \cdot (\Omega(\boldsymbol{\theta}_t) - \Omega(\boldsymbol{\theta}_{t-1})) \; ; \qquad (16)$$

     Step 2.3 : update and normalize weights: **for** $j = 1, 2, ..., n$,

$$w_{tj} \quad \leftarrow \quad w_{(t-1)j} \cdot \exp\left(-\alpha_t \pi_{j\iota(t)} + \delta_t\right)/Z_t \; ; \qquad (17)$$

**Return** $\boldsymbol{\theta}_T$;

---

with $\Omega$ a regularizer [Bach et al., 2011]. The following shows that when $f_e$ in eq. (11) is linear, there is a rado-loss equivalent to this regularized loss, *regardless* of $\Omega$.

**Theorem 7** *Suppose $\mathcal{H}$ contains linear classifiers. Let $(\ell_e(\mathcal{S}_e, \boldsymbol{\theta}), \ell_r(\mathcal{S}_r^*, \boldsymbol{\theta}))$ be any couple of equivalent example-rado losses such that $f_e$ in eq. (11) is linear:*

$$f_e(z) \quad = \quad a_e \cdot z + b_e \; , \qquad (14)$$

*for some $a_e > 0, b_e \in \mathbb{R}$. Then for any regularizer $\Omega(.)$ (assuming wlog $\Omega(\mathbf{0}) = 0$), the regularized example loss $\ell_e(\mathcal{S}_e, \boldsymbol{\theta}, \Omega)$ is equivalent to rado loss $\ell_r(\mathcal{S}_r^{*, \Omega, \boldsymbol{\theta}}, \boldsymbol{\theta})$ computed over **regularized** rados:*

$$\mathcal{S}_r^{*, \Omega, \boldsymbol{\theta}} \quad \doteq \quad \mathcal{S}_r^* \oplus \{-\tilde{\Omega}(\boldsymbol{\theta}) \cdot \boldsymbol{\theta}\} \; , \qquad (15)$$

*Here, $\oplus$ is Minkowski sum and $\tilde{\Omega}(\boldsymbol{\theta}) \doteq a_e \cdot \Omega(\boldsymbol{\theta})/\|\boldsymbol{\theta}\|_2^2$ if $\boldsymbol{\theta} \neq \mathbf{0}$ (and 0 otherwise).*

Theorem 7 applies to all rado losses (I-IV) in Table 1. The effect of regularization on rados is intuitive from the margin standpoint: assume that a "good" classifier $\boldsymbol{\theta}$ is one that ensures lowerbounded inner products $\boldsymbol{\theta}^\top \boldsymbol{z} \geq \tau$ for some *margin threshold* $\tau$. Then any good classifier on a regularized rado $\boldsymbol{\pi}_{\boldsymbol{\sigma}}$ shall actually meet, over *examples*, $\sum_{i:y_i=\sigma_i} \boldsymbol{\theta}^\top (y_i \cdot \boldsymbol{x}_i) \geq \tau + a_e \cdot \Omega(\boldsymbol{\theta})$. This inequality ties an "accuracy" of $\boldsymbol{\theta}$ (edges, left hand-side) and its sparsity (right-hand side). Clearly, Theorem 7 has an unfamiliar shape since regularisation modifies data in the rado world: a different $\boldsymbol{\theta}$, or a different $\Omega$, yields a different $\mathcal{S}_r^{*, \Omega, \boldsymbol{\theta}}$, and therefore it may seem very tricky to minimize such a regularized loss. Even more, iterative algorithms like boosting algorithms look at first glance a poor choice, since any update on $\boldsymbol{\theta}$ implies an update on the rados as well. What we show in the following Section is essentially the opposite for the exponential rado loss, and a generalization of the RADOBOOST algorithm of Nock et al. [2015], which does not modify rados, is a formal boosting algorithm for a broad set of regularizers. Also, remarkably, only the high-level code of the weak learner depends on the regularizer; that of the strong learner is not affected.

## 4  Boosting with (rado) regularized losses

$\Omega$-R.ADABOOST presents our approach to learning with rados regularized with regularizer $\Omega$ to minimise loss $\ell_r^{\exp}(\mathcal{S}_r, \boldsymbol{\theta}, \Omega)$ in eq. (45). Classifier $\boldsymbol{\theta}_t$ is defined as $\boldsymbol{\theta}_t \doteq \sum_{t'=1}^{t} \alpha_{\iota(t')} \cdot \mathbf{1}_{\iota(t')}$, where $\mathbf{1}_k$ is the $k^{th}$ canonical basis vector. The *expected edge* $r_t$ used to compute $\alpha_t$ in eq. (16) is based on the following basis assignation:

$$r_{\iota(t)} \quad \leftarrow \quad \frac{1}{\pi_{*\iota(t)}} \sum_{j=1}^{n} w_{tj} \pi_{j\iota(t)} \; (\in [-1, 1]) \; . \qquad (19)$$

The computation of $r_t$ is eventually tweaked by the weak learner, as displayed in Algorithm $\Omega$-WL. We investigate four choices for $\Omega$. For *each* of them, we prove the boosting ability of $\Omega$-R.ADABOOST ($\Gamma$ is symmetric positive definite, $S_d$ is the symmetric group of order $d$, $|\boldsymbol{\theta}|$ is the

**Algorithm 2** $\Omega$-WL, for $\Omega \in \{\|.\|_1, \|.\|_\Gamma^2, \|.\|_\infty, \|.\|_\Phi\}$

---

**Input** set of rados $\mathcal{S}_r \doteq \{\boldsymbol{\pi}_1, \boldsymbol{\pi}_2, ..., \boldsymbol{\pi}_n\}$; weights $\boldsymbol{w} \in \triangle_n$; parameters $\gamma \in (0,1)$, $\omega \in \mathbb{R}_+$; classifier $\boldsymbol{\theta} \in \mathbb{R}^d$;

*Step 1* : pick weak feature $\iota_* \in [d]$;

   Optional — use preference order: $\iota \succeq \iota' \Leftrightarrow |r_\iota| - \delta_\iota \geq |r_{\iota'}| - \delta_{\iota'}$

   // $\delta_\iota \doteq \omega \cdot (\Omega(\boldsymbol{\theta} + \alpha_\iota \cdot \mathbf{1}_\iota) - \Omega(\boldsymbol{\theta}))$, $r_\iota$ is given in (19) and $\alpha_\iota$ is given in (16)

*Step 2* : **if** $\Omega = \|.\|_\Gamma^2$ **then**

$$r_* \quad \leftarrow \quad \begin{cases} r_{\iota_*} & \text{if} \quad r_{\iota_*} \in [-\gamma, \gamma] \\ \text{sign}(r_{\iota_*}) \cdot \gamma & \text{otherwise} \end{cases} \quad ; \tag{18}$$

   **else** $r_* \leftarrow r_{\iota_*}$;

**Return** $(\iota_*, r_*)$;

---

vector whose coordinates are the absolute values of the coordinates of $\boldsymbol{\theta}$):

$$\Omega(\boldsymbol{\theta}) = \begin{cases} \|\boldsymbol{\theta}\|_1 & \doteq & |\boldsymbol{\theta}|^\top \mathbf{1} & \text{Lasso} \\ \|\boldsymbol{\theta}\|_\Gamma^2 & \doteq & \boldsymbol{\theta}^\top \Gamma \boldsymbol{\theta} & \text{Ridge} \\ \|\boldsymbol{\theta}\|_\infty & \doteq & \max_k |\theta_k| & \ell_\infty \\ \|\boldsymbol{\theta}\|_\Phi & \doteq & \max_{\mathbf{M} \in S_d} (\mathbf{M}|\boldsymbol{\theta}|)^\top \boldsymbol{\xi} & \text{SLOPE} \end{cases} \tag{20}$$

[Bach et al., 2011, Bogdan et al., 2015, Duchi and Singer, 2009, Su and Candès, 2015]. The coordinates of $\boldsymbol{\xi}$ in SLOPE are $\xi_k \doteq \Phi^{-1}(1 - kq/(2d))$ where $\Phi^{-1}(.)$ is the quantile of the standard normal distribution and $q \in (0,1)$; thus, the largest coordinates (in absolute value) of $\boldsymbol{\theta}$ are more penalized. We now establish the boosting ability of $\Omega$-R.ADABOOST. We give no direction for Step 1 in $\Omega$-WL, which is consistent with the definition of a weak learner in the boosting theory: all we require from the weak learner is $|r_.|$ no smaller than some weak learning threshold $\gamma_{\text{WL}} > 0$.

**Definition 8** *Fix any constant $\gamma_{\text{WL}} \in (0,1)$. $\Omega$-WL is said to be a $\gamma_{\text{WL}}$-Weak Learner iff the feature $\iota(t)$ it picks at iteration $t$ satisfies $|r_{\iota(t)}| \geq \gamma_{\text{WL}}$, for any $t = 1, 2, ..., T$.*

We also provide an optional step for the weak learner in $\Omega$-WL, which we exploit in the experimentations, which gives a total preference order on features to optimise further $\Omega$-R.ADABOOST.

**Theorem 9** *(boosting with ridge). Take $\Omega(.) = \|.\|_\Gamma^2$. Fix any $0 < a < 1/5$, and suppose that $\omega$ and the number of iterations $T$ of $\Omega$-R.ADABOOST are chosen so that*

$$\omega \quad < \quad (2a \min_k \max_j \pi_{jk}^2)/(T\lambda_\Gamma) , \tag{21}$$

*where $\lambda_\Gamma > 0$ is the largest eigenvalue of $\Gamma$. Then there exists some $\gamma > 0$ (depending on $a$, and given to $\Omega$-WL) such that for any fixed $0 < \gamma_{\text{WL}} < \gamma$, if $\Omega$-WL is a $\gamma_{\text{WL}}$-Weak Learner, then $\Omega$-R.ADABOOST returns at the end of the $T$ boosting iterations a classifier $\boldsymbol{\theta}_T$ which meets:*

$$\ell_r^{\exp}(\mathcal{S}_r, \boldsymbol{\theta}_T, \|.\|_\Gamma^2) \quad \leq \quad \exp(-a\gamma_{\text{WL}}^2 T/2) . \tag{22}$$

*Furthermore, if we fix $a = 1/7$, then we can fix $\gamma = 0.98$, and if $a = 1/10$, then we can fix $\gamma = 0.999$.*

Two remarks are in order. First, the cases $a = 1/7, 1/10$ show that $\Omega$-WL can still obtain large edges in eq. (19), so even a "strong" weak learner might fit in for $\Omega$-WL, without clamping edges. Second, the right-hand side of ineq. (21) may be very large if we consider that $\min_k \max_j \pi_{jk}^2$ may be proportional to $m^2$. So the constraint on $\omega$ is in fact loose.

**Theorem 10** *(boosting with lasso or $\ell_\infty$). Take $\Omega(.) \in \{\|.\|_1, \|.\|_\infty\}$. Suppose $\Omega$-WL is a $\gamma_{\text{WL}}$-Weak Learner for some $\gamma_{\text{WL}} > 0$. Suppose $\exists 0 < a < 3/11$ s. t. $\omega$ satisfies:*

$$\omega \quad = \quad a\gamma_{\text{WL}} \min_k \max_j |\pi_{jk}| . \tag{23}$$

*Then $\Omega$-R.ADABOOST returns at the end of the $T$ boosting iterations a classifier $\boldsymbol{\theta}_T$ which meets:*

$$\ell_r^{\exp}(\mathcal{S}_r, \boldsymbol{\theta}_T, \Omega) \quad \leq \quad \exp(-\tilde{T}\gamma_{\text{WL}}^2/2) , \tag{24}$$

*where $\tilde{T} = a\gamma_{\mathrm{WL}}T$ if $\Omega = \|.\|_1$, and $\tilde{T} = (T - T_*) + a\gamma_{\mathrm{WL}} \cdot T_*$ if $\Omega = \|.\|_\infty$; $T_*$ is the number of iterations where the feature computing the $\ell_\infty$ norm was updated[3].*

We finally investigate the SLOPE choice. The Theorem is proven for $\omega = 1$ in $\Omega$-R.ADABOOST, for two reasons: it matches the original definition [Bogdan et al., 2015] and furthermore it unveils an interesting connection between boosting and SLOPE properties.

**Theorem 11** *(boosting with* SLOPE*).* *Take* $\Omega(.) = \|.\|_\Phi$. *Let* $a \doteq \min\{3\gamma_{\mathrm{WL}}/11, \Phi^{-1}(1 - q/(2d))/\min_k \max_j |\pi_{jk}|\}$. *Suppose wlog* $|\theta_{Tk}| \geq |\theta_{T(k+1)}|, \forall k$, *and fix* $\omega = 1$. *Suppose (i)* $\Omega$-WL *is a* $\gamma_{\mathrm{WL}}$-*Weak Learner for some* $\gamma_{\mathrm{WL}} > 0$, *and (ii) the* $q$-*value is chosen to meet:*

$$q \geq 2 \cdot \max_k \left\{ \left(1 - \Phi\left(\frac{3\gamma_{\mathrm{WL}}}{11} \cdot \max_j |\pi_{jk}|\right)\right) \Big/ \left(\frac{k}{d}\right) \right\} \enspace .$$

*Then classifier* $\boldsymbol{\theta}_T$ *returned by* $\Omega$-R.ADABOOST *at the end of the* $T$ *boosting iterations satisfies:*

$$\ell_r^{\exp}(\mathcal{S}_r, \boldsymbol{\theta}_T, \|.\|_\Phi) \quad \leq \quad \exp(-a\gamma_{\mathrm{WL}}^2 T/2) \enspace . \tag{25}$$

Constraint (ii) on $q$ is interesting in the light of the properties of SLOPE [Su and Candès, 2015]. Modulo some assumptions, SLOPE yields a control the false discovery rate (FDR) — *i.e.*, negligible coefficients in the "true" linear model $\boldsymbol{\theta}^*$ that are found significant in the learned $\boldsymbol{\theta}$ —. Constraint (ii) links the "small" achievable FDR (upperbounded by $q$) to the "boostability" of the data: the fact that each feature $k$ can be chosen by the weak learner for a "large" $\gamma_{\mathrm{WL}}$, or has $\max_j |\pi_{jk}|$ large, precisely flags potential significant features, thus reducing the risk of sparsity errors, and allowing small $q$, which is constraint (ii). Using the second order approximation of normal quantiles [Su and Candès, 2015], a sufficient condition for (ii) is that, for some $K > 0$,

$$\gamma_{\mathrm{WL}} \min_j \max_j |\pi_{jk}| \quad \geq \quad K \cdot \sqrt{\log d + \log q^{-1}} \enspace ; \tag{26}$$

but $\min_j \max_j |\pi_{jk}|$ is proportional to $m$, so ineq. (26), and thus (ii), may hold even for small samples and $q$-values. An additional Theorem deferred to SM sor space considerations shows that for any applicable choice of regularization (eq. 20), the regularized log-loss of $\boldsymbol{\theta}_T$ *over examples* enjoys with high probability a monotonically decreasing upperbound with $T$ as: $\ell_e^{\log}(\mathcal{S}_e, \boldsymbol{\theta}, \Omega) \leq \log 2 - \kappa \cdot T + \tau(m)$, with $\tau(m) \to 0$ when $m \to \infty$ (and $\tau$ does not depend on $T$), and $\kappa > 0$ does not depend on $T$. Hence, $\Omega$-R.ADABOOST is an efficient proxy to boost the regularized log-loss over examples, using *whichever* of the ridge, lasso, $\ell_\infty$ or SLOPE regularization — establishing the first boosting algorithm for this choice —, or linear combinations of the choices, *e.g.* for elastic nets. If we were to compare Theorems 9 – 11 (eqs (22, 24, 25)), then the convergence looks best for ridge (the unsigned exponent is $\tilde{O}(\gamma_{\mathrm{WL}}^2)$) while it looks slightly worse for $\ell_\infty$ and SLOPE (the unsigned exponent is now $\tilde{O}(\gamma_{\mathrm{WL}}^3)$), the lasso being in between.

## 5 Experiments

We have implemented $\Omega$-WL[4] using the order suggested to retrieve the topmost feature in the order. Hence, the weak learner returns the feature maximising $|r_\iota| - \delta_\iota$. The rationale for this comes from the proofs of Theorems 9 — 11, showing that $\prod_t \exp(-(r_{\iota(t)}^2/2 - \delta_{\iota(t)}))$ is an upperbound on the exponential regularized rado-loss. We do not clamp the weak learner for $\Omega(.) = \|.\|_\Gamma^2$, so the weak learner is restricted to Step 1 in $\Omega$-WL[5].

The objective of these experiments is to evaluate $\Omega$-R.ADABOOST as a contender for supervised learning *per se*. We compared $\Omega$-R.ADABOOST to ADABOOST/$\ell_1$ regularized-ADABOOST [Schapire and Singer, 1999, Xi et al., 2009]. All algorithms are run for a total of $T = 1000$ iterations, and at the end of the iterations, the classifier in the sequence that minimizes the empirical loss is kept. Notice therefore that rado-based classifiers are evaluated on the training set which computes the

rados. To obtain very sparse solutions for regularized-ADABOOST, we pick its $\omega$ ($\beta$ in [Xi et al., 2009]) in $\{10^{-4}, 1, 10^4\}$. The complete results aggregate experiments on twenty (20) domains, all but one coming from the UCI [Bache and Lichman, 2013] (plus the Kaggle competition domain "Give me some credit"), with up to $d = $500+ features and $m = $100 000+ examples. Two tables, in the SM (Tables 1 and 2 in Section 3) report respectively the test errors and sparsity of classifiers, whose summary is given here in Table 2. The experimental setup is a ten-folds stratified cross validation for all algorithms and each domain. ADABOOST/regularized-ADABOOST is trained using the complete training fold. When the domain size $m \leq 40000$, the number of rados $n$ used for $\Omega$-R.ADABOOST is a random subset of rados of size equal to that of the training fold. When the domain size exceeds $40000$, a random set of $n = 10000$ rados is computed from the training fold. Thus, (i) there is no optimisation of the examples chosen to compute rados, (ii) we always keep a very small number of rados compared to the maximum available, and (iii) when the domain size gets large, we keep a comparatively tiny number of rados. Hence, the performances of $\Omega$-R.ADABOOST do not stem from any optimization in the choice or size of the rado sample.

| | Ada | $\emptyset$ | $\|\cdot\|^2_{I_d}$ | $\|\cdot\|_1$ | $\|\cdot\|_\infty$ | $\|\cdot\|_\Phi$ |
|---|---|---|---|---|---|---|
| Ada | | 11 | 10 | 10 | 8 | 9 |
| $\emptyset$ | 9 | | 3 | 3 | 2 | 1 |
| $\|\cdot\|^2_{I_d}$ | 10 | 17 | | 11 | 9 | 7 |
| $\|\cdot\|_1$ | 10 | 17 | 7 | | 7 | 4 |
| $\|\cdot\|_\infty$ | 11 | 18 | 9 | 9 | | 8 |
| $\|\cdot\|_\Phi$ | 10 | 19 | 10 | 10 | 11 | |

Table 2: Number of domains for which algorithm in *row* beats algorithm in *column* (Ada = **best** result of ADABOOST, $\emptyset$ = $\Omega$-R.ADABOOST not regularized, see text).

Experiments support several key observations. First, regularization consistently reduces the test error of $\Omega$-R.ADABOOST, by more than $15\%$ on Magic, and $20\%$ on Kaggle. In Table 2, $\Omega$-R.ADABOOST unregularized ("$\emptyset$") is virtually always beaten by its SLOPE regularized version. Second, $\Omega$-R.ADABOOST is able to obtain both very sparse *and* accurate classifiers (Magic, Hardware, Marketing, Kaggle). Third, $\Omega$-R.ADABOOST competes or beats ADABOOST on *all* domains, and is all the better as the domain gets bigger. Even qualitatively as seen in Table 2, the best result obtained by ADABOOST (regularized or not) does not manage to beat any of the regularized versions of $\Omega$-R.ADABOOST on the majority of the domains. Fourth, it is important to have several choices of regularizers at hand. On domain Statlog, the difference in test error between the worst and the best regularization of $\Omega$-R.ADABOOST exceeds $15\%$. Fifth, as already remarked [Nock et al., 2015], significantly subsampling rados (*e.g.* Marketing, Kaggle) still yields very accurate classifiers. Sixth, regularization in $\Omega$-R.ADABOOST successfully *reduces* sparsity to learn more accurate classifiers on several domains (Spectf, Transfusion, Hill-noise, Winered, Magic, Marketing), achieving efficient *adaptive* sparsity control. Last, the comparatively extremely poor results of ADABOOST on the biggest domains seems to come from another advantage of rados that the theory developed so far does not take into account: on domains for which some features are significantly correlated with the class and for which we have a large number of examples, the concentration of the expected feature value in rados seems to provide leveraging coefficients that tend to have much larger (absolute) value than in ADABOOST, making the convergence of $\Omega$-R.ADABOOST significantly faster than ADABOOST. For example, we have checked that it takes much more than the $T = 1000$ iterations for ADABOOST to start converging to the results of regularized $\Omega$-R.ADABOOST on Hardware or Kaggle.

## 6 Conclusion

We have shown that the recent equivalences between two example and rado losses can be unified and generalized via a principled representation of a loss function in a two-player zero-sum game. Furthermore, we have shown that this equivalence extends to regularized losses, where the regularization in the rado loss is performed over the rados themselves with Minkowski sums. Our theory and experiments on $\Omega$-R.ADABOOST with prominent regularizers (including ridge, lasso, $\ell_\infty$, SLOPE) indicate that when such a simple regularized form of the rado loss is available, it may help to devise accurate and efficient workarounds to boost a regularized loss over examples *via* the rado loss, even when the regularizer is significantly more involved like *e.g.* for group norms [Bach et al., 2011].

## Acknowledgments

Thanks are due to Stephen Hardy and Giorgio Patrini for stimulating discussions around this material.

## Footnotes

[1]Alternatively, $-\varphi_{\mathsf{e}}$ is permissible [Kearns and Mansour, 1999].

[2]To prevent notational overload, we blend notions of (pointwise) loss and (samplewise) risk, as just "losses".

[3]If several features match this criterion, $T_*$ is the total number of iterations for all these features.

[4]Code available at: http://users.cecs.anu.edu.au/~rnock/

[5]the values for $\omega$ that we test, in $\{10^{-u}, u \in \{0, 1, 2, 3, 4, 5\}\}$, are small with respect to the upperbound in ineq. (21) given the number of boosting steps ($T = 1000$), and would yield on most domains a maximal $\gamma \approx 1$.

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
