[Supplementary Material]

# On Regularizing Rademacher Observation Losses
## — Supplementary Material —

**Abstract**

This is the Supplementary Material to paper "On Regularizing Rademacher Observation Losses", by R. Nock, appearing in NIPS 2016.

# 1  Table of contents

# 2 Supplementary Material on Proofs

## 2.1 Proof of Theorem 2

We split the proof in two parts, the first concerning the case where both generators are differentiable since some of the derivations shall be used hereafter, and then the case where they are not. Remark that because of Lemma 4, we do not have to cover the case where just one of the two generators would be differentiable.

**Case 1**: $\varphi_e, \varphi_r$ are strictly convex and differentiable. We show in this case that being proportionate is equivalent to having:

$$\boldsymbol{p}^*(\boldsymbol{z}) \;\; = \;\; \mathrm{G}_m \boldsymbol{q}^*(\boldsymbol{z}) \;. \tag{1}$$

Solving eqs. (3) and (4) bring respectively:

$$p_i^*(\boldsymbol{z}) \;\; = \;\; {\varphi_e'}^{-1}\left(-\frac{1}{\mu_e} \cdot z_i\right) \;, \tag{2}$$

$$q_{\mathcal{J}}^*(\boldsymbol{z}) \;\; = \;\; {\varphi_r'}^{-1}\left(-\frac{1}{\mu_r} \cdot \sum_{i \in \mathcal{J}} z_i + \frac{\lambda}{\mu_r}\right) \;, \tag{3}$$

where $\lambda$ is picked so that $\boldsymbol{q}^*(\boldsymbol{z}) \in \mathbb{H}^{2^m}$, that is,

$$\sum_{\mathcal{J} \subseteq [m]} {\varphi_r'}^{-1}\left(-\frac{1}{\mu_r} \cdot \sum_{i \in \mathcal{J}} z_i + \frac{\lambda}{\mu_r}\right) \;\; = \;\; 1 \;. \tag{4}$$

We obtain

$$\mathcal{L}_e(\boldsymbol{z}) \;\; = \;\; -\mu_e \sum_{i \in [m]} \varphi_e^\star\left(-\frac{1}{\mu_e} \cdot z_i\right) \;, \tag{5}$$

$$\mathcal{L}_r(\boldsymbol{z}) \;\; = \;\; \lambda - \mu_r \sum_{\mathcal{J} \subseteq [m]} \varphi_r^\star\left(-\frac{1}{\mu_r} \cdot \sum_{i \in \mathcal{J}} z_i + \frac{\lambda}{\mu_r}\right) \;, \tag{6}$$

where $\varphi^\star(z) \doteq \sup_{z'}\{zz' - \varphi(z')\}$ denotes the convex conjugate of $\varphi$. It follows from eq. (5) that:

$$\begin{aligned}
\frac{\partial}{\partial z_i} \mathcal{L}_e(\boldsymbol{z}) \;\; &= \;\; {\varphi_e^\star}'\left(-\frac{1}{\mu_e} \cdot z_i\right) \\
&= \;\; {\varphi_e'}^{-1}\left(-\frac{1}{\mu_e} \cdot z_i\right) \tag{7} \\
&= \;\; p_i^*(\boldsymbol{z}) \;, \tag{8}
\end{aligned}$$

where eq. (7) follows from properties of $\varphi^\star$. We also have

$$
\begin{aligned}
\frac{\partial}{\partial z_i} & \mathcal{L}_r(z) \\
= {} & \left( 1 - \sum_{\mathcal{I} \subseteq [m]} \varphi_r'^{-1} \left( -\frac{1}{\mu_r} \cdot \sum_{j \in \mathcal{I}} z_j + \frac{\lambda}{\mu_r} \right) \right) \cdot \frac{\partial \lambda}{\partial z_i} \\
& + \sum_{\mathcal{I} \subseteq [m]: i \in \mathcal{I}} \varphi_r'^{-1} \left( -\frac{1}{\mu_r} \cdot \sum_{j \in \mathcal{I}} z_j + \frac{\lambda}{\mu_r} \right) \\
= {} & \frac{\partial \lambda}{\partial z_i} \\
& + \sum_{\mathcal{I} \subseteq [m]} \left( 1_{i \in \mathcal{I}} - \frac{\partial \lambda}{\partial z_i} \right) \varphi_r'^{-1} \left( -\frac{1}{\mu_r} \cdot \sum_{j \in \mathcal{I}} z_j + \frac{\lambda}{\mu_r} \right) \\
= {} & \frac{\partial \lambda}{\partial z_i} + \sum_{\mathcal{I} \subseteq [m]} \left( 1_{i \in \mathcal{I}} - \frac{\partial \lambda}{\partial z_i} \right) \cdot q_{\mathcal{I}}^*(z) \\
= {} & \frac{\partial \lambda}{\partial z_i} \cdot \left( 1 - \sum_{\mathcal{I} \subseteq [m]} q_{\mathcal{I}}^*(z) \right) + \sum_{\mathcal{I} \subseteq [m]} 1_{i \in \mathcal{I}} \cdot q_{\mathcal{I}}^*(z) \\
= {} & \sum_{\mathcal{I} \subseteq [m]} 1_{i \in \mathcal{I}} \cdot q_{\mathcal{I}}^*(z) \;,
\end{aligned}
\tag{9}
$$

since $q^*(z) \in \mathbb{H}^{2^m}$.

Now suppose $\varphi_e$ and $\varphi_r$ proportionate. It comes that there exists $(\mu_e, \mu_r)$ such that the gradients of eq. (7) yield $\nabla \mathcal{L}_e(z) = \nabla \mathcal{L}_r(z)$, and from eqs. (8) and (9) we obtain $p^*(z) = G_m q^*(z)$.

Reciprocally, having $p^*(z) = G_m q^*(z)$ for some $\varphi_e, \varphi_r$ and $\mu_e, \mu_r > 0$ implies as well $\nabla \mathcal{L}_e(z) = \nabla \mathcal{L}_r(z)$ from eqs. (8) and (9), and therefore eq. (7) holds as well. This ends the proof of Case 1 for Theorem 2.

**Case 2**: $\varphi_e, \varphi_r$ are not differentiable. To simplify the statement and proofs, we assume that $\boldsymbol{\mu_e} = \boldsymbol{\mu_r} = 1$. We define the following problems

$$
\mathcal{L}_e(z) \; \doteq \; \inf_{p \in \mathbb{R}^m} z^\top p + \varphi_e(p) \;,
\tag{10}
$$

$$
\mathcal{L}_r(z) \; \doteq \; \inf_{q \in \mathbb{H}^{2^m}} z^\top G_m q + \varphi_r(q) \;,
\tag{11}
$$

where $\varphi_e : \mathbb{R}^m \to \mathbb{R}$ and $\varphi_r : \mathbb{R}^{2^m} \to \mathbb{R}$ are convex. Recall that $\partial \mathcal{L}_e$ and $\partial \mathcal{L}_r$ are their subdifferentials, and $p(z)$ and $q(z)$ the arguments of the infima, assuming without loss of generality that they are finite. We now show that being proportionate is equivalent to having, for any $z$,

$$
p(z) \; \in \; \partial \mathcal{L}_r(z) \;,
\tag{12}
$$

$$
G_m q(z) \; \in \; \partial \mathcal{L}_e(z) \;.
\tag{13}
$$

This property is an immediate consequence of the following property, which we shall in fact show:

$$
\begin{aligned}
\boldsymbol{p}(\boldsymbol{z}) &\in& \partial\mathcal{L}_{\mathsf{e}}(\boldsymbol{z}) \;, & \quad (14)\\
\mathrm{G}_m\boldsymbol{q}(\boldsymbol{z}) &\in& \partial\mathcal{L}_{\mathsf{r}}(\boldsymbol{z}) \;. & \quad (15)
\end{aligned}
$$

Granted all (12—15) hold, Eq. (1) of Theorem 2 follows whenever subgradients are singletons. To see why the statement of the Theorem follows from (12–13), if the functions are proportionate, then their subdifferentials match from Definition 1 (main file) and we immediately get (12) and (13) from (14) and (15). If, on the other hand, we have both (12) and (13), then we get from (14) and (15) that $\partial\mathcal{L}_{\mathsf{e}}(\boldsymbol{z}) \cap \partial\mathcal{L}_{\mathsf{r}}(\boldsymbol{z}) \neq \emptyset, \forall \boldsymbol{z}$ and so $\boldsymbol{0} \in \partial(\mathcal{L}_{\mathsf{e}}(\boldsymbol{z}) - \mathcal{L}_{\mathsf{r}}(\boldsymbol{z}))$, yieding the fact that the epigraphs of $\mathcal{L}_{\mathsf{e}}(\boldsymbol{z})$ and $\mathcal{L}_{\mathsf{r}}(\boldsymbol{z})$ match by a translation of some $b$ that does not depend on $\boldsymbol{z}$, and by extension, the fact that $\varphi_{\mathsf{e}}$ and $\varphi_{\mathsf{r}}$ meet Definition 1 (main file) and are proportionate.

To show (14), we first remark that $-\boldsymbol{z}' \in \partial\varphi_{\mathsf{e}}(\boldsymbol{p}(\boldsymbol{z}'))$ for any $\boldsymbol{z}'$ because of the definition of $\boldsymbol{p}$ in (10). So, from the definition of subdifferentials, for any $\boldsymbol{z}$,

$$
\varphi_{\mathsf{e}}(\boldsymbol{p}(\boldsymbol{z}')) + (-\boldsymbol{z}')^{\top}(\boldsymbol{p}(\boldsymbol{z}) - \boldsymbol{p}(\boldsymbol{z}')) \;\leq\; \varphi_{\mathsf{e}}(\boldsymbol{p}(\boldsymbol{z})) \;.
$$

Reorganising and substracting $\boldsymbol{z}^{\top}\boldsymbol{p}(\boldsymbol{z})$ to both sides, we get

$$
\begin{aligned}
&-\varphi_{\mathsf{e}}(\boldsymbol{p}(\boldsymbol{z}')) - \boldsymbol{z}'^{\top}\boldsymbol{p}(\boldsymbol{z}')\\
&\geq\; -\varphi_{\mathsf{e}}(\boldsymbol{p}(\boldsymbol{z})) - \boldsymbol{z}^{\top}\boldsymbol{p}(\boldsymbol{z}) + (-\boldsymbol{p}(\boldsymbol{z}))^{\top}(\boldsymbol{z}' - \boldsymbol{z}) \;,
\end{aligned}
$$

which shows that $-\boldsymbol{p}(\boldsymbol{z}) \in \partial - (\varphi_{\mathsf{e}}(\boldsymbol{p}(\boldsymbol{z})) + \boldsymbol{z}^{\top}\boldsymbol{p}(\boldsymbol{z}))$, and so $\boldsymbol{p}(\boldsymbol{z}) \in \partial\mathcal{L}_{\mathsf{e}}(\boldsymbol{z})$.

We then tackle (15). We show that there exists $\lambda \in \mathbb{R}$ such that $\lambda \cdot \boldsymbol{1}_{2^m} - \mathrm{G}_m^{\top}\boldsymbol{z} \in \partial\varphi_{\mathsf{r}}(\boldsymbol{q}(\boldsymbol{z}))$ at the optimal $\boldsymbol{q}(\boldsymbol{z})$. Suppose it is not the case. Then because of the definition of subgradients, for any $\lambda \in \mathbb{R}$, there exists $\boldsymbol{q} \in \mathbb{H}^{2^m}, \boldsymbol{q} \neq \boldsymbol{q}(\boldsymbol{z})$ such that

$$
\varphi_{\mathsf{r}}(\boldsymbol{q}(\boldsymbol{z})) + (\lambda \cdot \boldsymbol{1}_{2^m} - \mathrm{G}_m^{\top}\boldsymbol{z})^{\top}(\boldsymbol{q} - \boldsymbol{q}(\boldsymbol{z})) \;>\; \varphi_{\mathsf{r}}(\boldsymbol{q}) \;.
$$

Reorganising and using the fact that $\boldsymbol{q}, \boldsymbol{q}_* \in \mathbb{H}^{2^m}$, we get $\varphi_{\mathsf{r}}(\boldsymbol{q}(\boldsymbol{z})) + \boldsymbol{z}^{\top}\mathrm{G}_m\boldsymbol{q}(\boldsymbol{z}) > \varphi_{\mathsf{r}}(\boldsymbol{q}) + \boldsymbol{z}^{\top}\mathrm{G}_m\boldsymbol{q}$, contradicting the optimality of $\boldsymbol{q}(\boldsymbol{z})$. Consider any $\boldsymbol{z}'$ and its corresponding optimal $\boldsymbol{q}(\boldsymbol{z}')$. Since $\lambda' \cdot \boldsymbol{1}_{2^m} - \mathrm{G}_m^{\top}\boldsymbol{z} \in \partial\varphi_{\mathsf{r}}(\boldsymbol{q}(\boldsymbol{z}))$ for some $\lambda' \in \mathbb{R}$, we get from the definition of subgradients that

$$
\begin{aligned}
&\varphi_{\mathsf{r}}(\boldsymbol{q}(\boldsymbol{z}))\\
&\geq\; \varphi_{\mathsf{r}}(\boldsymbol{q}(\boldsymbol{z}')) + (\lambda' \cdot \boldsymbol{1}_{2^m} - \mathrm{G}_m^{\top}\boldsymbol{z}')^{\top}(\boldsymbol{q}(\boldsymbol{z}) - \boldsymbol{q}(\boldsymbol{z}')) \;.
\end{aligned}
$$

Reorganising and using the fact that $\boldsymbol{q}(\boldsymbol{z}), \boldsymbol{q}(\boldsymbol{z}') \in \mathbb{H}^{2^m}$, we get

$$
\begin{aligned}
&-(\varphi_{\mathsf{r}}(\boldsymbol{q}(\boldsymbol{z}')) + \boldsymbol{z}'^{\top}\mathrm{G}_m\boldsymbol{q}(\boldsymbol{z}'))\\
&\geq\; -(\varphi_{\mathsf{r}}(\boldsymbol{q}(\boldsymbol{z})) + \boldsymbol{z}^{\top}\mathrm{G}_m\boldsymbol{q}(\boldsymbol{z}))\\
&\quad + (-\mathrm{G}_m\boldsymbol{q}(\boldsymbol{z}))^{\top}(\boldsymbol{z}' - \boldsymbol{z}) \;, \qquad\qquad (16)
\end{aligned}
$$

showing that $-\mathrm{G}_m\boldsymbol{q}(\boldsymbol{z}) \in \partial - (\varphi_{\mathsf{r}}(\boldsymbol{q}(\boldsymbol{z})) + \boldsymbol{z}^{\top}\mathrm{G}_m\boldsymbol{q}(\boldsymbol{z}))$, and so $\mathrm{G}_m\boldsymbol{q}(\boldsymbol{z}) \in \partial\mathcal{L}_{\mathsf{r}}(\boldsymbol{z})$.

## 2.2 Proof of Lemma 4

Take $m = 1$, and replace $\boldsymbol{z}$ by real $z_1$. We have $L_e(p, z_1) = pz_1 + \varphi_e(z_1)$ and $L_r(\boldsymbol{q}, z) = q_{\{1\}}z_1 + \varphi_r(q_{\{1\}}) + \varphi_r(q_\emptyset)$. Remark that we can drop the constraint $\boldsymbol{q} \in \mathbb{H}^2$ since then $q_\emptyset = 1 - q_{\{1\}}$. So we get

$$
\begin{aligned}
\mathcal{L}_r(\boldsymbol{q}) &= \min_{q \in \mathbb{R}} qz_1 + \mu_r\varphi_r(q) + \mu_r\varphi_r(1-q) \\
&= \min_{q \in \mathbb{R}} qz_1 + \mu_r\varphi_{s(r)}(q) \\
&= -\mu_r\varphi_{s(r)}^\star\left(-\frac{1}{\mu_r} \cdot z_1\right) ,
\end{aligned}
$$

whereas

$$
\mathcal{L}_e(p) = -\mu_e\varphi_r^\star\left(-\frac{1}{\mu_e} \cdot z_1\right) ,
$$

and since $\varphi_e$ and $\varphi_r$ are proportionate, then

$$
\varphi_r^\star\left(-\frac{1}{\mu_e} \cdot z_1\right) = \frac{\mu_r}{\mu_e} \cdot \varphi_{s(r)}^\star\left(-\frac{1}{\mu_r} \cdot z_1\right) - \frac{b}{\mu_e} . \tag{17}
$$

We then make the variable change $z \doteq -z_1/\mu_e$ and get

$$
\varphi_e^\star(z) = \frac{\mu_r}{\mu_e} \cdot \varphi_{s(r)}^\star\left(\frac{\mu_e}{\mu_r} \cdot z\right) - \frac{b}{\mu_e} , \tag{18}
$$

which yields, since $\varphi_e$, $\varphi_r$, and by extension $\varphi_{s(r)}$, are all convex and lower-semicontinuous,

$$
\varphi_e(z) = \frac{\mu_r}{\mu_e} \cdot \varphi_{s(r)}(z) + \frac{b}{\mu_e} , \tag{19}
$$

as claimed.

## 2.3 Proof of Theorem 6

We detail all proofs for all entries in Table 1 (see main file). Hereafter, we just write $\varphi_s$ instead of $\varphi_{s(r)}$.

**Lemma 1** $\varphi_r(z) \doteq z \log z - z$ *is proportionate to* $\varphi_e \doteq \varphi_s = z \log z + (1-z) \log(1-z) - 1$, *whenever* $\mu_e = \mu_r$.

**Proof** We use the fact that whenever $\varphi$ is differentiable, $\varphi^\star(z) \doteq z \cdot \varphi'^{-1}(z) - \varphi(\varphi'^{-1}(z))$. We have $\varphi_r'(z) = \log z$, $\varphi_r'^{-1}(z) = \exp z = \varphi_r^\star(z)$. Therefore, the Lagrange multiplier $\lambda$ in (4) is

$$
\lambda = -\mu_r \cdot \log\left(\sum_{\mathcal{J} \subseteq [m]} \exp\left(-\frac{1}{\mu_r} \cdot \sum_{i \in \mathcal{J}} z_i\right)\right) , \tag{20}
$$

which yields from (3):

$$
q_{\mathcal{J}}^*(\boldsymbol{z}) = \frac{\exp\left(-\frac{1}{\mu_r} \cdot \sum_{i \in \mathcal{J}} z_i\right)}{\sum_{\mathcal{J} \subseteq [m]} \exp\left(-\frac{1}{\mu_r} \cdot \sum_{j \in \mathcal{J}} z_j\right)} , \forall \mathcal{J} \subseteq [m] .
$$

On the other hand, we also have $\varphi_s'(z) = \log(z/(1-z))$, $\varphi_s'^{-1}(z) = \exp(z)/(1 + \exp(z))$ and $\varphi_s^\star(z) = 1 + \log(1 + \exp(z))$, which yields from (2):

$$p_i^*(z) = \frac{\exp\left(-\frac{1}{\mu_e} \cdot z_i\right)}{1 + \exp\left(-\frac{1}{\mu_e} \cdot z_i\right)} \ , \forall i \in [m] \ . \tag{21}$$

We then check that for any $i \in [m]$, we indeed have

$$\sum_{\mathcal{I} \subseteq [m]} 1_{i \in \mathcal{I}} \cdot q_{\mathcal{I}}^*(z)$$

$$= \sum_{\mathcal{I} \subseteq [m]} 1_{i \in \mathcal{I}} \cdot \frac{\exp\left(-\frac{1}{\mu_r} \cdot \sum_{i' \in \mathcal{I}} z_{i'}\right)}{\sum_{\mathcal{J} \subseteq [m]} \exp\left(-\frac{1}{\mu_r} \cdot \sum_{j \in \mathcal{J}} z_j\right)}$$

$$= \exp\left(-\frac{1}{\mu_e} \cdot z_i\right) \cdot \frac{\sum_{\mathcal{J} \subseteq [m] \setminus \{i\}} \exp\left(-\frac{1}{\mu_r} \cdot \sum_{j \in \mathcal{J}} z_j\right)}{\sum_{\mathcal{J} \subseteq [m]} \exp\left(-\frac{1}{\mu_r} \cdot \sum_{j \in \mathcal{J}} z_j\right)}$$

$$= \exp\left(-\frac{1}{\mu_e} \cdot z_i\right) \cdot \frac{c}{\left(1 + \exp\left(-\frac{1}{\mu_e} \cdot z_i\right)\right) \cdot c}$$

$$= \frac{\exp\left(-\frac{1}{\mu_r} \cdot z_i\right)}{1 + \exp\left(-\frac{1}{\mu_r} \cdot z_i\right)} \ , \tag{22}$$

with $c \doteq \sum_{\mathcal{J} \subseteq [m] \setminus \{i\}} \exp\left(-\frac{1}{\mu_r} \cdot \sum_{j \in \mathcal{J}} z_j\right)$. We check that eq. (22) equals eq. (21) whenever $\mu_e = \mu_r$. Hence eq. (1) holds. We conclude that $\varphi_r$ and $\varphi_e = \varphi_s$ are proportionate whenever $\mu_e = \mu_r$ (End of the proof of Lemma 1). ∎

**Corollary 2** *The following example and rado losses are equivalent for any* $\mu > 0$*:*

$$\ell_e(z, \mu) = \sum_{i \in [m]} \log\left(1 + \exp\left(-\frac{1}{\mu} \cdot z_i\right)\right) \ , \tag{23}$$

$$\ell_r(z, \mu) = \sum_{\mathcal{I} \subseteq [m]} \exp\left(-\frac{1}{\mu} \cdot \sum_{i \in \mathcal{I}} z_i\right) \ . \tag{24}$$

**Proof** Consider $\varphi_r(z) \doteq z \log z - z$ and $\varphi_e = \varphi_s$. We obtain from eq. (5):

$$-\mathcal{L}_e(z)$$

$$= f_e\left(\sum_{i \in [m]} \log\left(1 + \exp\left(-\frac{1}{\mu_e} \cdot z_i\right)\right)\right) \ ,$$

with $f_\text{e}(z) = \mu_\text{e} \cdot z + \mu_\text{e} m$. We have also $\varphi_\text{r}^\star(z) = \exp(z)$, and so using $\lambda$ in eq. (20) and eq. (6), we obtain

$$
\begin{aligned}
&-\mathcal{L}_\text{r}(\boldsymbol{z}) \\
&= \mu_\text{r} \cdot \log\left(\sum_{\mathcal{J} \subseteq [m]} \exp\left(-\frac{1}{\mu_\text{r}} \cdot \sum_{i \in \mathcal{J}} z_i\right)\right) \\
&\quad + \mu_\text{r} \cdot \exp\left(\frac{\lambda}{\mu_\text{r}}\right) \cdot \sum_{\mathcal{J} \subseteq [m]} \exp\left(-\frac{1}{\mu_\text{r}} \cdot \sum_{i \in \mathcal{J}} z_i\right) \\
&= \mu_\text{r} \cdot \log\left(\sum_{\mathcal{J} \subseteq [m]} \exp\left(-\frac{1}{\mu_\text{r}} \cdot \sum_{i \in \mathcal{J}} z_i\right)\right) \\
&\quad + \mu_\text{r} \cdot \underbrace{\frac{\sum_{\mathcal{J} \subseteq [m]} \exp\left(-\frac{1}{\mu_\text{r}} \cdot \sum_{i \in \mathcal{J}} z_i\right)}{\sum_{\mathcal{J} \subseteq [m]} \exp\left(-\frac{1}{\mu_\text{r}} \cdot \sum_{i \in \mathcal{J}} z_i\right)}}_{=1} \\
&= f_\text{r}\left(\sum_{\mathcal{J} \subseteq [m]} \exp\left(-\frac{1}{\mu_\text{r}} \cdot \sum_{i \in \mathcal{J}} z_i\right)\right) \quad,
\end{aligned}
$$

with $f_\text{r}(z) = \mu_\text{r} \cdot \log z + \mu_\text{r}$. We get from Lemma 1 that the following example and rado risks are equivalent whenever $\mu_\text{e} = \mu_\text{r}$:

$$
\ell_\text{e}(\boldsymbol{z}, \mu_\text{e}) = \sum_{i \in [m]} \log\left(1 + \exp\left(-\frac{1}{\mu_\text{e}} \cdot z_i\right)\right) \quad, \tag{25}
$$

$$
\ell_\text{r}(\boldsymbol{z}, \mu_\text{r}) = \sum_{\mathcal{J} \subseteq [m]} \exp\left(-\frac{1}{\mu_\text{r}} \cdot \sum_{i \in \mathcal{J}} z_i\right) \quad, \tag{26}
$$

from which we get the statement of the Corollary by fixing $\mu = \mu_\text{e} = \mu_\text{r}$ (end of the proof of Corollary 2). ∎

**Lemma 3** $\varphi_r(z) \doteq (1/2) \cdot z^2$ *is proportionate to* $\varphi_e \doteq \varphi_s = (1/2) \cdot (1 - 2z(1 - z))$ *whenever* $\mu_e = \mu_r / 2^{m-1}$.

**Proof** We proceed as in the proof of Lemma 1. We have $\varphi_\text{r}'(z) = z$, $\varphi_\text{r}'^{-1}(z) = z$ and $\varphi_\text{r}^\star(z) = \varphi_\text{r}(z)$. Therefore, the Lagrange multiplier $\lambda$ in (4) is

$$
\lambda = \frac{\mu_\text{r}}{2^m} + \frac{1}{2^m} \cdot \sum_{\mathcal{J} \subseteq [m]} \sum_{i \in \mathcal{J}} z_i \tag{27}
$$

$$
= \frac{\mu_\text{r}}{2^m} + \frac{1}{2} \cdot \sum_{i \in [m]} z_i \quad, \tag{28}
$$

since any $i$ belongs exactly to half of the subsets of $[m]$. We obtain:

$$q_J^*(z) = \frac{1}{2^m} - \frac{1}{\mu_r} \cdot \sum_{i \in J} z_i + \frac{1}{2\mu_r} \cdot \sum_{i \in [m]} z_i \ , \forall J \subseteq [m] \ .$$

On the other hand, we also have $\varphi_s'(z) = 2z - 1$, ${\varphi_s'}^{-1}(z) = (1 + z)/2$ and $\varphi_s^\star(z) = -(1/4) + (1/4) \cdot (1 + z)^2$, which yields from (2):

$$p_i^*(z) = \frac{1}{2} \cdot \left( 1 - \frac{1}{\mu_e} \cdot z_i \right) \ , \forall i \in [m] \ . \tag{29}$$

We then check that for any $i \in [m]$, we have

$$\sum_{J \subseteq [m]} 1_{i \in J} \cdot q_J^*(z)$$

$$= \sum_{J \subseteq [m]} 1_{i \in J} \cdot \left( \frac{1}{2^m} - \frac{1}{\mu_r} \cdot \sum_{i \in J} z_i + \frac{1}{2\mu_r} \cdot \sum_{i \in [m]} z_i \right)$$

$$= \frac{1}{2} - \frac{1}{\mu_r} \cdot \sum_{J \subseteq [m]} 1_{i \in J} \cdot \sum_{i \in J} z_i + \frac{2^{m-2}}{\mu_r} \cdot \sum_{i \in [m]} z_i$$

$$= \frac{1}{2} - \frac{2^{m-1}}{\mu_r} \cdot z_i - \frac{1}{\mu_r} \cdot \sum_{J \subseteq [m] \setminus \{i\}} \sum_{i \in J} z_i$$

$$+ \frac{2^{m-2}}{\mu_r} \cdot \sum_{i \in [m]} z_i$$

$$= \frac{1}{2} - \frac{2^{m-1}}{\mu_r} \cdot z_i - \frac{2^{m-2}}{\mu_r} \cdot \sum_{i \in [m] \setminus \{i\}} z_i$$

$$+ \frac{2^{m-2}}{\mu_r} \cdot \sum_{i \in [m]} z_i$$

$$= \frac{1}{2} - \frac{2^{m-1}}{\mu_r} \cdot z_i + \frac{2^{m-2}}{\mu_r} \cdot z_i$$

$$= \frac{1}{2} \left( 1 - \frac{2^{m-1}}{\mu_r} \cdot z_i \right) \ . \tag{30}$$

We check that eq. (30) equals eq. (29) whenever $\mu_e = \mu_r / 2^{m-1}$. Hence eq. (1) holds. We conclude that $\varphi_r$ is proportionate to $\varphi_e = \varphi_s$ whenever $\mu_e = \mu_r / 2^{m-1}$ (end of the proof of Lemma 3). ∎

**Corollary 4** *The following example and rado losses are equivalent, for any $\mu > 0$:*

$$\ell_e(z, \mu) = \sum_{i \in [m]} \left( 1 - \frac{1}{\mu} \cdot z_i \right)^2 \ , \tag{31}$$

$$\ell_r(z, \mu) = - \left( \mathbb{E}_J \left[ \frac{1}{\mu} \cdot \sum_{i \in J} z_i \right] - \mu \cdot \mathbb{V}_J \left[ \frac{1}{\mu} \cdot \sum_{i \in J} z_i \right] \right) \ , \tag{32}$$

*where $\mathbb{E}_{\mathcal{I}}[X(\mathcal{I})]$ and $\mathbb{V}_{\mathcal{I}}[X(\mathcal{I})]$ denote the expectation and variance of $X$ wrt uniform weights on $\mathcal{I} \subseteq$* $[m]$.

**Proof** Consider $\varphi_{\mathsf{r}}(z) \doteq (1/2) \cdot z^2$ and $\varphi_{\mathsf{e}} = \varphi_{\mathsf{s}}$. We obtain from eq. (5):

$$
\begin{aligned}
&-\mathcal{L}_{\mathsf{e}}(\boldsymbol{z}) \\
&= \quad f_{\mathsf{e}}\left(\sum_{i \in [m]} \left(1 - \frac{1}{\mu_{\mathsf{e}}} \cdot z_i\right)^2\right) \quad ,
\end{aligned}
$$

with $f_\mathsf{e}(z) = (\mu_\mathsf{e}/4) \cdot z + (\mu_\mathsf{e} m/4)$. We have also $\varphi_\mathsf{r}^\star(z) = (1/2) \cdot z^2$, and so using eq. (6) and $\lambda$ in eq. (27), we obtain

$$
\begin{aligned}
-\mathcal{L}_\mathsf{r}(\boldsymbol{z}) \\
= & -\frac{\mu_\mathsf{r}}{2^m} - \frac{1}{2^m} \cdot \sum_{\mathcal{I} \subseteq [m]} \sum_{i \in \mathcal{I}} z_i \\
& + \frac{1}{2\mu_\mathsf{r}} \sum_{\mathcal{I} \subseteq [m]} \left( \sum_{i \in \mathcal{I}} z_i - \frac{\mu_\mathsf{r}}{2^m} - \frac{1}{2^m} \cdot \sum_{\mathcal{I} \subseteq [m]} \sum_{i \in \mathcal{I}} z_i \right)^2 \\
= & -\frac{\mu_\mathsf{r}}{2^m} - \frac{1}{2^m} \cdot \sum_{\mathcal{I} \subseteq [m]} \sum_{i \in \mathcal{I}} z_i + \frac{\mu_\mathsf{r}}{2^{m+1}} \\
& - \frac{1}{2^m} \cdot \underbrace{\sum_{\mathcal{I} \subseteq [m]} \left( \sum_{i \in \mathcal{I}} z_i - \frac{1}{2^m} \cdot \sum_{\mathcal{I} \subseteq [m]} \sum_{i \in \mathcal{I}} z_i \right)}_{=0} \\
& + \frac{1}{2\mu_\mathsf{r}} \sum_{\mathcal{I} \subseteq [m]} \left( \sum_{i \in \mathcal{I}} z_i - \frac{1}{2^m} \cdot \sum_{\mathcal{I} \subseteq [m]} \sum_{i \in \mathcal{I}} z_i \right)^2 \\
= & -\frac{\mu_\mathsf{r}}{2^{m+1}} - \frac{1}{2^m} \cdot \sum_{\mathcal{I} \subseteq [m]} \sum_{i \in \mathcal{I}} z_i \\
& + \frac{2^{m-1}}{\mu_\mathsf{r}} \cdot \frac{1}{2^m} \cdot \sum_{\mathcal{I} \subseteq [m]} \left( \sum_{i \in \mathcal{I}} z_i - \frac{1}{2^m} \cdot \sum_{\mathcal{I} \subseteq [m]} \sum_{i \in \mathcal{I}} z_i \right)^2 \\
= & -\frac{\mu_\mathsf{r}}{2^{m+1}} \\
& - \mathbb{E}_{\mathcal{I} \sim [m]} \left[ \sum_{i \in \mathcal{I}} z_i \right] + \frac{2^{m-1}}{\mu_\mathsf{r}} \cdot \mathbb{V}_{\mathcal{I} \sim [m]} \left[ \sum_{i \in \mathcal{I}} z_i \right] \\
= & -\frac{\mu_\mathsf{r}}{2^{m+1}} \\
& + \frac{\mu_\mathsf{r}}{2^{m-1}} \cdot \left( - \left( \begin{array}{c} \mathbb{E}_{\mathcal{I} \sim [m]} \left[ \frac{2^{m-1}}{\mu_\mathsf{r}} \cdot \sum_{i \in \mathcal{I}} z_i \right] \\ - \frac{\mu_\mathsf{r}}{2^{m-1}} \cdot \mathbb{V}_{\mathcal{I} \sim [m]} \left[ \frac{2^{m-1}}{\mu_\mathsf{r}} \sum_{i \in \mathcal{I}} z_i \right] \end{array} \right) \right) \\
= & \; f_\mathsf{r} \left( - \left( \begin{array}{c} \mathbb{E}_{\mathcal{I} \sim [m]} \left[ \frac{2^{m-1}}{\mu_\mathsf{r}} \cdot \sum_{i \in \mathcal{I}} z_i \right] \\ - \frac{\mu_\mathsf{r}}{2^{m-1}} \cdot \mathbb{V}_{\mathcal{I} \sim [m]} \left[ \frac{2^{m-1}}{\mu_\mathsf{r}} \sum_{i \in \mathcal{I}} z_i \right] \end{array} \right) \right) \;,
\end{aligned}
\tag{33}
$$

with $f_r(z) = (\mu_r/2^{m-1}) \cdot z - (\mu_r/2^{m+1})$. Therefore, it comes from Lemma 3 that the following example and rado risks are equivalent whenever $\mu_e = \mu_r/2^{m-1}$:

$$\ell_e(\boldsymbol{z}, \mu_e) = \sum_{i\in[m]} \left(1 - \frac{1}{\mu_e} \cdot z_i\right)^2 \;,$$

$$\ell_r(\boldsymbol{z}, \mu_r) = -\left(\mathbb{E}_{\mathfrak{I}}\left[\frac{2^{m-1}}{\mu_r} \cdot \sum_{i\in\mathfrak{I}} z_i\right]\right.$$
$$\left. - \frac{\mu_r}{2^{m-1}} \cdot \mathbb{V}_{\mathfrak{I}}\left[\frac{2^{m-1}}{\mu_r} \cdot \sum_{i\in\mathfrak{I}} z_i\right]\right) \;.$$

There remains to fix $\mu \doteq \mu_e = \mu_r/2^{m-1}$ to obtain the statement of the Corollary (end of the proof of Corollary 4). ∎

We now investigate cases of non differentiable proportionate generators, the first of which is self-proportionate ($\varphi_e = \varphi_r$). We let $\chi_{\mathcal{A}}(z)$ be the indicator function: $\chi_{\mathcal{A}}(z) \doteq 0$ if $z \in \mathcal{A}$ (and $+\infty$ otherwise), convex since $\mathcal{A} = [0,1]$ is convex.

**Lemma 5** $\varphi_r(z) \doteq \chi_{[0,1]}(z)$ *is self-proportionate,* $\forall \mu_e, \mu_r$.

**Proof** Define $\triangle_d$ as the $d$-dimensional probability simplex. Then it comes with that choice of $\varphi_r(q_{\mathfrak{I}})$:

$$\min_{\boldsymbol{q}\in\mathbb{H}^{2^m}} L_r(\boldsymbol{q}, \boldsymbol{z})$$
$$= \min_{\boldsymbol{q}\in\triangle_{2^m}} \sum_{\mathfrak{I}\subseteq[m]} q_{\mathfrak{I}} \sum_{i\in\mathfrak{I}} z_i$$
$$= \begin{cases} 0 & \text{if } \sum_{i\in\mathfrak{I}} z_i > 0, \forall \mathfrak{I} \neq \emptyset \;, \\ \sum_{i:z_i<0} z_i & \text{otherwise} \end{cases} \;, \tag{34}$$

since whenever no $z_i$ is negative, the minimum is achieved by putting all the mass (1) on $q_\emptyset$, and when some are negative, the minimum is achieved by putting all the mass on the smallest over all $\mathfrak{I}$ of $\sum_{i\in\mathfrak{I}} z_i$, which is the one which collects all the indexes of the negative coordinates in $\boldsymbol{z}$.

On the other hand, remark that fixing $\varphi_e \doteq \varphi_s$ still yields $\varphi_e(z) = \chi_{[0,1]}(z) = \varphi_r(z)$, yet this time we have the following on $L_e(\boldsymbol{p}, \boldsymbol{z})$:

$$\min_{\boldsymbol{p}\in\mathbb{R}^m} L_r(\boldsymbol{q}, \boldsymbol{z}) = \min_{\boldsymbol{p}\in[0,1]^m} \sum_{i\in[m]} p_i z_i$$
$$= -\mu_e \cdot \sum_{i\in[m]} \max\left\{0, -\frac{1}{\mu_e} \cdot z_i\right\} \;, \tag{35}$$

since the optimal choice for $p_i^*$ is to put 1 only when $z_i$ is negative. We obtain $\boldsymbol{p}^*(\boldsymbol{z}) = \mathrm{G}_m \boldsymbol{q}^*(\boldsymbol{z})$ for any choice of $\mu_e, \mu_r$, and so $\varphi_r(z)$ is self-proportionate for any $\mu_e, \mu_r$. This ends the proof of Lemma 5. ∎

**Corollary 6** *The following example and rado losses are equivalent, for any $\mu_e$, $\mu_r$:*

$$\ell_e(\boldsymbol{z}, \mu_e) \;=\; \sum_{i \in [m]} \max \left\{ 0, -\frac{1}{\mu_e} \cdot z_i \right\} \;, \tag{36}$$

$$\ell_r(\boldsymbol{z}, \mu_r) \;=\; \max \left\{ 0, \max_{\mathcal{I} \subseteq [m]} \left\{ -\frac{1}{\mu_r} \cdot \sum_{i \in \mathcal{I}} z_i \right\} \right\} \;. \tag{37}$$

**Proof** We obtain from Lemma 5 that $-\mathcal{L}_r(\boldsymbol{z}) = f_r(\ell_r(\boldsymbol{z}, \mu_r))$ with $f_r(z) = \mu_r \cdot z$ and:

$$\ell_r(\boldsymbol{z}, \mu_r) \;=\; \max \left\{ 0, \max_{\mathcal{I} \subseteq [m]} \left\{ -\frac{1}{\mu_r} \cdot \sum_{i \in \mathcal{I}} z_i \right\} \right\} \;. \tag{38}$$

On the other hand, it comes from eq. (35) that $-\mathcal{L}_e(\boldsymbol{z}) = f_r(\ell_e(\boldsymbol{z}, \mu_e))$ with $f_e(z) = \mu_e \cdot z$ and:

$$\ell_e(\boldsymbol{z}, \mu_e) \;=\; \sum_{i \in [m]} \max \left\{ 0, -\frac{1}{\mu_e} \cdot z_i \right\} \;. \tag{39}$$

This concludes the proof of Corollary 6. ∎

**Lemma 7** $\varphi_r(z) \doteq \chi_{\left[\frac{1}{2^m}, \frac{1}{2}\right]}(z)$ *is proportionate to* $\varphi_e \doteq \varphi_s = \chi_{\left\{\frac{1}{2}\right\}}(z)$, *for any $\mu_e$, $\mu_r$.*

**Proof** The choice of

$$\varphi_r(z) \;=\; \chi_{\left[\frac{1}{2^m}, \frac{1}{2}\right]}(z) \;, \tag{40}$$

under the constraint that $\boldsymbol{q} \in \mathbb{H}^{2^m}$, enforces $q_{\mathcal{I}}^* = 1/2^m, \forall \mathcal{I} \subseteq [m]$. Furthermore, fixing $\varphi_e \doteq \varphi_s$ indeed yields

$$\begin{aligned}
\varphi_e &= \chi_{\left[\frac{1}{2^m}, \frac{1}{2}\right]}(z) + \chi_{\left[\frac{1}{2^m}, \frac{1}{2}\right]}(1 - z) \\
&= \chi_{\left\{\frac{1}{2}\right\}}(z) \;,
\end{aligned} \tag{41}$$

which enforces $p_i^* = 1/2$, $\forall i$. Since each $i$ belongs to exactly $2^{m-1}$ subsets of $[m]$, we obtain $\boldsymbol{p}^*(\boldsymbol{z}) = \mathrm{G}_m \boldsymbol{q}^*(\boldsymbol{z})$, for any $\mu_e$, $\mu_r$, and so $\varphi_r$ is proportionate to $\varphi_e = \varphi_s$ for any $\mu_e$, $\mu_r$. This concludes the proof of Lemma 7. ∎

**Corollary 8** *The following example and rado losses are equivalent, for any $\mu_e$, $\mu_r$:*

$$\ell_e(\boldsymbol{z}, \mu_e) \;=\; \sum_i -\frac{1}{\mu_e} \cdot z_i \;, \tag{42}$$

$$\ell_r(\boldsymbol{z}, \mu_r) \;=\; \mathbb{E}_{\mathcal{I}} \left[ -\frac{1}{\mu_r} \cdot \sum_{i \in \mathcal{I}} z_i \right] \;. \tag{43}$$

**Proof** We obtain from Lemma 7 that $-\mathcal{L}_\mathsf{r}(\boldsymbol{z}) = f_\mathsf{r}(\ell_\mathsf{r}(\boldsymbol{z}, \mu_\mathsf{r}))$ with $f_\mathsf{r}(z) = z$ and:

$$\ell_\mathsf{r}(\boldsymbol{z}, \mu_\mathsf{r}) \quad = \quad \mathbb{E}_\mathfrak{I}\left[-\frac{1}{\mu_\mathsf{r}} \cdot \sum_{i \in \mathfrak{I}} z_i\right] \quad .$$

On the other hand, it comes from eq. (35) that $-\mathcal{L}_\mathsf{e}(\boldsymbol{z}) = f_\mathsf{r}(\ell_\mathsf{e}(\boldsymbol{z}, \mu_\mathsf{e}))$ with $f_\mathsf{e}(z) = (1/2) \cdot z$ and:

$$\ell_\mathsf{e}(\boldsymbol{z}, \mu_\mathsf{e}) \quad = \quad \sum_i -\frac{1}{\mu_\mathsf{e}} \cdot z_i \quad .$$

This concludes the proof of Corollary 8. $\blacksquare$

## 2.4 Proof of Theorem 7

The key to the poof is the constraint $\boldsymbol{q} \in \mathbb{H}^m$ in eq. (4). Since $f_\mathsf{e}(z) = a_\mathsf{e} \cdot z + b_\mathsf{e}$, we have $\mathcal{L}_\mathsf{e}(\boldsymbol{z}) = a_\mathsf{e} \cdot (\ell_\mathsf{e}(\boldsymbol{z}) + \omega) + b_\mathsf{e} - a_\mathsf{e} \cdot \omega$ for any $\omega \in \mathbb{R}$. It follows from eq. (7) (see main file) that $a_\mathsf{e} \cdot (\ell_\mathsf{e}(\boldsymbol{z}) + \omega) + b_\mathsf{e} - a_\mathsf{e} \cdot \omega = \mathcal{L}_\mathsf{r}(\boldsymbol{z}) + b = \sum_{\mathfrak{I} \subseteq [m]} q_\mathfrak{I}^* \sum_{i \in \mathfrak{I}} z_i + \mu_\mathsf{r} \sum_{\mathfrak{I} \subseteq [m]} \varphi_\mathsf{r}(q_\mathfrak{I}^*) + b$, and so

$$a_\mathsf{e} \cdot (\ell_\mathsf{e}(\boldsymbol{z}) + \omega) + b_\mathsf{e}$$
$$= \quad -\left\{\min_{\boldsymbol{q} \in \mathbb{H}^m}\left(\sum_{\mathfrak{I} \subseteq [m]} q_\mathfrak{I} \sum_{i \in \mathfrak{I}} z_i + \mu_\mathsf{r} \sum_{\mathfrak{I} \subseteq [m]} \varphi_\mathsf{r}(q_\mathfrak{I})\right) - a_\mathsf{e}\omega\right\}$$
$$+ b$$
$$= \quad -\min_{\boldsymbol{q} \in \mathbb{H}^m}\left(\sum_{\mathfrak{I} \subseteq [m]} q_\mathfrak{I}\left(\sum_{i \in \mathfrak{I}} z_i - a_\mathsf{e}\omega\right) + \mu_\mathsf{r} \sum_{\mathfrak{I} \subseteq [m]} \varphi_\mathsf{r}(q_\mathfrak{I})\right)$$
$$+ b \quad ,$$

since $\boldsymbol{q} \in \mathbb{H}^m$ and $a_\mathsf{e}, \omega, a$ are not a function of $\boldsymbol{q}$. We thus get $a_\mathsf{e} \cdot (\ell_\mathsf{e}(\boldsymbol{z}) + \omega) + b_\mathsf{e} = a_\mathsf{r} \cdot f_\mathsf{r}\left(\tilde{\ell}_\mathsf{r}(\boldsymbol{z})\right) + b_\mathsf{r}$, where $\tilde{\ell}_\mathsf{r}(\boldsymbol{z})$ equals $\ell_\mathsf{r}(\boldsymbol{z})$ in which each $\sum_{i \in \mathfrak{I}} z_i$ is replaced by $\sum_{i \in \mathfrak{I}} z_i - a_\mathsf{e}\omega$. For $z_i = \boldsymbol{\theta}^\top(y_i \cdot \boldsymbol{x}_i)$ and $\omega = \Omega(\boldsymbol{\theta})$, we obtain that whenever $\boldsymbol{\theta} \neq \boldsymbol{0}$, $\forall \mathfrak{I} \subseteq [m]$,

$$\sum_{i \in \mathfrak{I}} z_i + a_\mathsf{e}\omega \quad = \quad \boldsymbol{\theta}^\top\left(\pi_{\boldsymbol{\sigma}} - \frac{a_\mathsf{e}\Omega(\boldsymbol{\theta})}{\|\boldsymbol{\theta}\|_2^2} \cdot \boldsymbol{\theta}\right) \quad , \tag{44}$$

for $\sigma_i = y_i$ iff $i \in \mathfrak{I}$ (and $-y_i$ otherwise), and the statement of the Theorem follows.

**Remark** — one important question, not addressed in the main file to save space, is the way the minimisation of the regularized rado loss impacts the minimisation of the regularized examples loss when one *subsamples* the rados, and learns $\boldsymbol{\theta}$ from some $\mathcal{S}_\mathsf{r} \subseteq \mathcal{S}_\mathsf{r}^*$ with eventually $|\mathcal{S}_\mathsf{r}| \ll |\mathcal{S}_\mathsf{r}^*|$. We

give an answer for the log-loss [Nock et al., 2015] (row I in Table 1), and for this objective define the $\Omega$-regularized exp-rado-loss computed over $\mathcal{S}_r$, with $|\mathcal{S}_r| = n$ and $\omega > 0$ user-fixed:

$$
\begin{aligned}
&\ell_r^{\exp}(\mathcal{S}_r, \boldsymbol{\theta}, \Omega) \\
&\doteq \frac{1}{n} \cdot \sum_{j \in [n]} \exp\left( -\boldsymbol{\theta}^\top \left( \boldsymbol{\pi}_j - \omega \cdot \frac{\Omega(\boldsymbol{\theta})}{\|\boldsymbol{\theta}\|_2^2} \cdot \boldsymbol{\theta} \right) \right) \;,
\end{aligned}
\tag{45}
$$

whenever $\boldsymbol{\theta} \neq \mathbf{0}$ (otherwise, we discard the factor depending on $\omega$ in the formula). We assume that $\Omega$ is a norm, and let $\ell_r^{\exp}(\mathcal{S}_r, \boldsymbol{\theta})$ denote the unregularized loss ($\omega = 0$ in eq. (45)), and we let $\ell_e^{\log}(\mathcal{S}_e, \boldsymbol{\theta}, \Omega) \doteq (1/m) \sum_i \log\left(1 + \exp\left(-\boldsymbol{\theta}^\top(y_i \cdot \boldsymbol{x}_i)\right)\right) + \Omega(\boldsymbol{\theta})$ denote the $\Omega$-regularized log-loss. Notice that we normalize losses. We define the open ball $\mathcal{B}_\Omega(\mathbf{0}, r) \doteq \{\boldsymbol{x} \in \mathbb{R}^d : \Omega(\boldsymbol{x}) < r\}$ and $r_\pi^\star \doteq (1/m) \cdot \max_{\mathcal{S}_r^*} \Omega^\star(\boldsymbol{\pi}_\sigma)$, where $\Omega^\star$ is the dual norm of $\Omega$. The following Theorem is a direct application of Theorem 3 in [Nock et al., 2015], and shows mild conditions on $\mathcal{S}_r \subseteq \mathcal{S}_r^*$ for the minimization of $\ell_r^{\exp}(\mathcal{S}_r, \boldsymbol{\theta}, \Omega)$ to indeed yield that of $\ell_e^{\log}(\mathcal{S}_e, \boldsymbol{\theta}, \Omega)$.

**Theorem 9** *Assume $\Theta \subseteq \mathcal{B}_{\|.\|_2}(\mathbf{0}, r_\theta)$, with $r_\theta > 0$. Let $\varrho(\boldsymbol{\theta}) \doteq (\sup_{\boldsymbol{\theta}' \in \Theta} \max_{\boldsymbol{\pi}_\sigma \in \mathcal{S}_r^*} \exp(-\boldsymbol{\theta}'^\top \boldsymbol{\pi}_\sigma))/\ell_r^{\exp}(\mathcal{S}_r^*, \boldsymbol{\theta})$. Then if $m$ is sufficiently large, $\forall \delta > 0$, there is probability $\geq 1 - \delta$ over the sampling of $\mathcal{S}_r$ that any $\boldsymbol{\theta} \in \Theta$ satisfies:*

$$
\begin{aligned}
\ell_e^{\log}(\mathcal{S}_e, \boldsymbol{\theta}, \Omega) &\leq \log 2 + (1/m) \cdot \log \ell_r^{\exp}(\mathcal{S}_r, \boldsymbol{\theta}, \Omega) \\
&+ O\left( \frac{\varrho(\boldsymbol{\theta})}{m^\beta} \cdot \sqrt{\frac{r_\theta r_\pi^\star}{n} + \frac{d}{nm} \log \frac{n}{d\delta}} \right) \;,
\end{aligned}
$$

*as long as $\omega \geq um$ for some constant $u > 0$.*

## 2.5   Proof of Theorem 9

The proof of the Theorem contains two parts, the first of which follows ADABOOST's exponential convergence rate proof, and the second departs from this proof to cover $\Omega$-R.ADABOOST.

We use the fact that $\alpha_{\iota(t)} \pi_{j\iota(t)} = \alpha_{\iota(t)} \cdot \mathbf{1}_{\iota(t)}^\top \boldsymbol{\pi}_j = (\boldsymbol{\theta}_T - \boldsymbol{\theta}_{T-1})^\top \boldsymbol{\pi}_j$ to unravel the weights as:

$$
\begin{aligned}
w_{Tj} \\
&= \frac{w_{(T-1)j}}{Z_T} \cdot \exp\left( -\alpha_{\iota(T)} \pi_{j\iota(T)} + \delta_T \right) \\
&= \frac{w_{(T-1)j}}{Z_T} \cdot \exp\left( \begin{array}{l} -(\boldsymbol{\theta}_T - \boldsymbol{\theta}_{T-1})^\top \boldsymbol{\pi}_j \\ +\omega \cdot (\|\boldsymbol{\theta}_T\|_2^2 - \|\boldsymbol{\theta}_{T-1}\|_2^2) \end{array} \right) \\
&= \frac{w_{(T-1)j}}{Z_T} \cdot \exp\left( \begin{array}{l} -\boldsymbol{\theta}_T^\top (\boldsymbol{\pi}_j - \omega \cdot \boldsymbol{\theta}_T) \\ +\boldsymbol{\theta}_{T-1}^\top (\boldsymbol{\pi}_j - \omega \cdot \boldsymbol{\theta}_{T-1}) \end{array} \right) \\
&= \frac{w_0}{\prod_{t=1}^T Z_t} \cdot \exp\left( \begin{array}{l} -\boldsymbol{\theta}_T^\top (\boldsymbol{\pi}_j - \omega \cdot \boldsymbol{\theta}_T) \\ +\boldsymbol{\theta}_0^\top (\boldsymbol{\pi}_j - \omega \cdot \boldsymbol{\theta}_0) \end{array} \right) \\
&= \frac{w_0}{\prod_{t=1}^T Z_t} \cdot \exp\left( -\boldsymbol{\theta}_T^\top (\boldsymbol{\pi}_j - \omega \cdot \boldsymbol{\theta}_T) \right) \;,
\end{aligned}
\tag{46}
$$
$$
\tag{47}
$$

since the sums telescope in eq. (46) when we unravel the weight update and $\boldsymbol{\theta}_0 = \mathbf{0}$. We therefore get

$$
\ell_r^{\exp}(\mathcal{S}_r, \boldsymbol{\theta}, \|.\|_2^2) = \prod_{t=1}^T Z_t \;,
\tag{48}
$$

as in the classical ADABOOST analysis [Schapire and Singer, 1999]. This time however, we have, letting $\tilde{\pi}_{j\iota(t)} \doteq \pi_{j\iota(t)}/\pi_{*\iota(t)} \in [-1, 1]$ and $\tilde{\alpha}_{\iota(t)} \doteq \pi_{*\iota(t)} \cdot \alpha_t$ for short,

$$
\begin{aligned}
Z_{t+1} &= \sum_{j\in[n]} w_{tj} \cdot \exp\left(-\alpha_{\iota(t)}\pi_{j\iota(t)} + \delta_t\right) \\
&= \exp(\delta_t) \cdot \sum_{j\in[n]} w_{tj} \cdot \exp\left(-\alpha_{\iota(t)}\pi_{j\iota(t)}\right) \\
&= \exp(\delta_t) \cdot \sum_{j\in[n]} w_{tj} \cdot \exp\left(-\tilde{\alpha}_{\iota(t)}\tilde{\pi}_{j\iota(t)}\right) \\
&\leq \frac{\exp(\delta_t)}{2} \\
&\quad \cdot \sum_{j\in[n]} w_{tj} \cdot \left( \begin{array}{c} (1+\tilde{\pi}_{j\iota(t)}) \cdot \exp\left(-\tilde{\alpha}_{\iota(t)}\right) \\ +(1-\tilde{\pi}_{j\iota(t)}) \cdot \exp\left(\tilde{\alpha}_{\iota(t)}\right) \end{array} \right) \\
&= \exp(\delta_t) \cdot \sqrt{1-r_t^2} \\
&= \exp\left( \omega \cdot (\|\boldsymbol{\theta}_t\|_2^2 - \|\boldsymbol{\theta}_{t-1}\|_2^2) - \frac{1}{2}\ln\frac{1}{1-r_t^2} \right) \ .
\end{aligned}
$$
(49)
(50)

This is where our proof follows a different path from ADABOOST's: in eq. (50), we do not upperbound the $\sqrt{1-r_t^2}$ term, so it can absorb more easily the new $\exp(\delta_t)$ factor which appears because of regularization.

Ineq. (49) holds because of the convexity of $\exp$, and eq. (50) is an equality when $r_t < \gamma$. If $r_t > \gamma$ is clamped to $r_t \leftarrow \gamma$ by the weak learner in (18), then we have instead the derivation

$$
\begin{aligned}
\sum_{j\in[n]} w_{tj} \cdot &\left( \begin{array}{c} (1+\tilde{\pi}_{j\iota(t)}) \cdot \exp\left(-\tilde{\alpha}_{\iota(t)}\right) \\ +(1-\tilde{\pi}_{j\iota(t)}) \cdot \exp\left(\tilde{\alpha}_{\iota(t)}\right) \end{array} \right) \\
&= (1+r_t) \cdot \sqrt{\frac{1-\gamma}{1+\gamma}} + (1-r_t) \cdot \sqrt{\frac{1+\gamma}{1-\gamma}} \\
&\leq 2\sqrt{1-\gamma^2} \ ,
\end{aligned}
$$
(51)

since function in (51) is decreasing on $r_t > 0$. If $r_t < -\gamma$ is clamped to $r_t \leftarrow -\gamma$, we get the same conclusion as in ineq (51) because this time $\tilde{\alpha}_{\iota(t)} = (1/2) \cdot \ln((1-\gamma)/(1+\gamma))$. Summarising, whether $r_t$ has been clamped or not by the weak learner in (18), we get

$$
\begin{aligned}
Z_{t+1} &\leq \exp\left( \omega \cdot (\|\boldsymbol{\theta}_t\|_2^2 - \|\boldsymbol{\theta}_{t-1}\|_2^2) - \frac{1}{2}\ln\frac{1}{1-r_t^2} \right) \ ,
\end{aligned}
$$
(52)

with the additional fact that $|r_t| \leq \gamma$. For any feature index $k \in [d]$, let $\mathcal{F}_k \subseteq [T]$ the iteration indexes

for which $\iota(t) = k$. Letting $\lambda_\Gamma \ (> 0)$ the largest eigenvalue of $\Gamma$, we obtain:

$$\prod_{t=1}^{T} Z_t$$

$$\leq \ \exp\left(\omega \cdot \|\boldsymbol{\theta}_T\|_\Gamma^2 - \sum_t \frac{1}{2} \log \frac{1}{1 - r_t^2}\right)$$

$$\leq \ \exp\left(\omega \lambda_\Gamma \cdot \|\boldsymbol{\theta}_T\|_2^2 - \sum_t \frac{1}{2} \log \frac{1}{1 - r_t^2}\right)$$

$$= \exp\left(-\frac{1}{2} \cdot \sum_{k \in [d]} \Lambda_k\right) \ , \tag{53}$$

With

$$\begin{aligned} \Lambda_k \ \doteq \ & \log \frac{1}{\prod_{t:\iota(t)\in\mathcal{F}_k}(1 - r_t^2)} \\ & -\frac{\omega\lambda_\Gamma}{2\pi_{*k}^2} \log^2 \prod_{t:\iota(t)\in\mathcal{F}_k} \left(\frac{1 + r_t}{1 - r_t}\right) \ . \end{aligned} \tag{54}$$

Since $(\sum_{l=1}^{u} a_l)^2 \leq u \sum_{l=1}^{u} a_l^2$ and $\min_k \max_j |\pi_{jk}| \leq |\pi_{*k}|$, $\Lambda_k$ satisfies:

$$\begin{aligned} \Lambda_k \ \geq \ & \sum_{t:\iota(t)\in\mathcal{F}_k} \left\{\log \frac{1}{1 - r_t^2}\right. \\ & \left. -\frac{T_k \omega \lambda_\Gamma}{2M^2} \log^2 \frac{1 + r_t}{1 - r_t}\right\} \ , \end{aligned} \tag{55}$$

with $T_k \doteq |\mathcal{F}_k|$ and $M \doteq \min_k \max_j |\pi_{jk}|$. For any $a > 0$, let

$$f_a(z) \ \doteq \ \frac{1}{az^2} \cdot \left(\log \frac{1}{1 - z^2} - a \cdot \log^2 \frac{1 + z}{1 - z}\right) - 1 \ .$$

It satisfies

$$\begin{aligned} f_a(z) \ \approx_0 \ & \left(\frac{1}{a} - 5\right) + \left(\frac{1}{2a} - \frac{8}{3}\right) \cdot z^2 \\ & + \left(\frac{1}{3a} - \frac{92}{45}\right) \cdot z^4 + o(z^4) \ . \end{aligned} \tag{56}$$

Since $f_a(z)$ is continuous for any $a \neq 0$, $\forall 0 < a < 1/5$, $\exists z_*(a) > 0$ such that $f_a(z) \geq 0, \forall z \in [0, z_*]$. So, for any such $a < 1/5$ and any $\omega$ satisfying $\omega < (2aM^2)/(T_k\lambda_\Gamma)$, as long as each $r_t \leq z_*(a)$, we shall obtain

$$\Lambda_k \ \geq \ a \sum_{t:\iota(t)\in\mathcal{F}_k} r_t^2 \ . \tag{57}$$

There remains to tune $\gamma \leq z_*(a)$, and remark that if we fix $a = 1/7$, then numerical calculations reveal that $z_*(a) > 0.98$, and if $a = 1/10$ then numerical calculations give $z_*(a) > 0.999$, completing the statement of Theorem 9.

## 2.6 Proof of Theorem 10

We consider the case $\Omega(.) = \|.\|_\infty$, from which we shall derive the case $\Omega(.) = \|.\|_1$. We proceed as in the proof of Theorem 9, with the main change that we have now $\delta_t = \omega \cdot (\|\boldsymbol{\theta}_t\|_\infty - \|\boldsymbol{\theta}_{t-1}\|_\infty)$, so in place of $\Lambda_k$ in ineq. (53) we have to use, letting $k_*$ any feature that gives the $\ell_\infty$ norm,

$$\Lambda_k \;\doteq\; \begin{cases} \sum_{t:\iota(t)\in\mathcal{F}_k} \log \frac{1}{1-r_t^2} \\ -\frac{\omega}{\pi_{*k}} \left| \sum_{t:\iota(t)\in\mathcal{F}_k} \log \frac{1+r_t}{1-r_t} \right| & \text{if} \quad k = k_* \\ \sum_{t:\iota(t)\in\mathcal{F}_k} \log \frac{1}{1-r_t^2} & \text{otherwise} \end{cases} \tag{58}$$

It also comes

$$\begin{aligned} \Lambda_{k_*} \\ \geq\; & \sum_{t:\iota(t)\in\mathcal{F}_{k_*}} \left\{ \log \frac{1}{1-r_t^2} - \frac{\omega}{\pi_{*k_*}} \log \frac{1+|r_t|}{1-|r_t|} \right\} \\ \geq\; & \sum_{t:\iota(t)\in\mathcal{F}_{k_*}} \left\{ \log \frac{1}{1-r_t^2} - \frac{\omega}{M} \log \frac{1+|r_t|}{1-|r_t|} \right\} \;, \end{aligned} \tag{59}$$

with $M \doteq \min_k \max_j |\pi_{jk}|$. Let us analyze $\Lambda_{k_*}$ and define for any $b > 0$

$$\begin{aligned} g_b(z) \;\doteq\; & \log \frac{1}{1-z^2} - b \cdot \log \frac{1+z}{1-z} \\ & - \left( -2bz + z^2 - \frac{2bz^3}{3} \right) \;. \end{aligned} \tag{60}$$

Inspecting $g_b$ shows that $g_b(0) = 0$, $g_b'(0) = 0$ and $g_b(z)$ is convex over $[0,1)$ for any $b \leq 3$, which shows that $g_b(z) \geq 0, \forall z \in [0,1), \forall b \leq 3$, and so, after dividing by $bz^2$ and reorganising, yields in these cases:

$$\begin{aligned} & \frac{1}{bz^2} \cdot \left( \log \frac{1}{1-z^2} - b \cdot \log \frac{1+z}{1-z} \right) - 1 \\ \geq\; & \left( -\frac{2}{z} + \left( \frac{1}{b} - 1 \right) - \frac{2z}{3} \right) \;. \end{aligned} \tag{61}$$

Hence, both functions being continuous on $(0,1)$, the function in the left-hand side zeroes before the one in the right-hand side (when this one does on $(0,1)$). The zeroes of the polynomial

$$p_b(z) \;\doteq\; -\frac{2z^2}{3} + \left( \frac{1}{b} - 1 \right) z - 2 \tag{62}$$

exist iff $b \leq \sqrt{3}/(4+\sqrt{3})$, in which case any $z \in [0,1)$ must satisfy

$$z \;\geq\; \frac{3}{4} \cdot \left( \frac{1}{b} - 1 - \sqrt{\left( \frac{1}{b} - 1 \right)^2 - \frac{16}{3}} \right) \tag{63}$$

to guarantee that $p_b(z) \geq 0$. Whenever this happens, we shall have from (61):

$$\log \frac{1}{1-z^2} - b \cdot \log \frac{1+z}{1-z} \; \geq \; bz^2 \; . \tag{64}$$

Since $\Omega$-WL is a $\gamma_{\mathrm{WL}}$-weak learner, if we can guarantee that the right-hand side of ineq. (63) is no more than $\gamma_{\mathrm{WL}}$, then there is nothing more to require from the weak learner to have ineq. (64) — and therefore to have $\Lambda_{k_*} \geq b\gamma_{\mathrm{WL}}^2 \cdot |\mathcal{F}_{k_*}|$. This yields equivalently the following constraint on $b$:

$$b \; \leq \; \frac{\frac{8\gamma_{\mathrm{WL}}}{3}}{\frac{16\gamma_{\mathrm{WL}}^2}{9} + \frac{8\gamma_{\mathrm{WL}}}{3} + \frac{16}{3}} \; . \tag{65}$$

Since $\gamma_{\mathrm{WL}} \leq 1$, ineq (65) ensured as long as

$$b \; \leq \; \frac{\frac{8\gamma_{\mathrm{WL}}}{3}}{\frac{16}{9} + \frac{8}{3} + \frac{16}{3}} = \frac{3\gamma_{\mathrm{WL}}}{11} \; , \tag{66}$$

which also guarantees $b \leq \sqrt{3}/(4+\sqrt{3})$. So, letting $T_* \doteq |\mathcal{F}_{k_*}|$ and recollecting

$$b \; \doteq \; \frac{\omega}{\min_k \max_j |\pi_{jk}|} \tag{67}$$

from eq. (59), we obtain from ineqs (59) and (64):

$$\Lambda_{k_*} \; \geq \; \frac{\omega T_* \gamma_{\mathrm{WL}}^2}{\min_k \max_j |\pi_{jk}|} \; . \tag{68}$$

We need to ensure $\omega \leq 3\min_k \max_j |\pi_{jk}|\gamma_{\mathrm{WL}}/11$ from ineq . (66), which holds if we pick it according to eq. (23). In this case, we finally obtain

$$\Lambda_{k_*} \; \geq \; (a\gamma_{\mathrm{WL}}T_*) \cdot \gamma_{\mathrm{WL}}^2 \; . \tag{69}$$

Now, since $\log(1/(1-x^2)) \geq x^2$, we also have for $k \neq k_*$ in eq. (58),

$$\begin{aligned}
\Lambda_k \; &= \; \sum_{t:\iota(t)\in\mathcal{F}_k} \log \frac{1}{1-r_t^2} \\
&\geq \; \sum_{t:\iota(t)\in\mathcal{F}_k} r_t^2 \\
&\geq \; |\mathcal{F}_k|\gamma_{\mathrm{WL}}^2 \; , \forall k \neq k_* \; . 
\end{aligned} \tag{70}$$

So, we finally obtain from eq. (51) and ineq. (53),

$$\ell_r^{\exp}(\mathcal{S}_r, \boldsymbol{\theta}, \|.\|_2^2) \; \leq \; \exp\left(-\frac{\tilde{T}\gamma_{\mathrm{WL}}^2}{2}\right) \; , \tag{71}$$

with $\tilde{T} \doteq (T - T_*) + a\gamma_{\mathrm{WL}} \cdot T_*$, as claimed when $\Omega(.) = \|.\|_\infty$. The case $\Omega = \|.\|_1$ follows form the fact that all $\Lambda_k$ match the bound of $\Lambda_{k_*}$.

## 2.7 Proof of Theorem 11

We use the proof of Theorem 10, since when $\Omega(.) = \|.\|_\Phi$, eq. (58) becomes

$$\Lambda_k \doteq \sum_{t:\iota(t)\in\mathcal{F}_k} \log \frac{1}{1-r_t^2} \tag{72}$$

$$- \frac{\xi_k}{\pi_{*k}} \left| \sum_{t:\iota(t)\in\mathcal{F}_k} \log \frac{1+r_t}{1-r_t} \right|$$

$$\geq \sum_{t:\iota(t)\in\mathcal{F}_k} \left\{ \log \frac{1}{1-r_t^2} - \frac{\xi_k}{\max_j |\pi_{jk}|} \log \frac{1+|r_t|}{1-|r_t|} \right\} \quad , \tag{73}$$

assuming without loss of generality that the classifier at iteration $T$, $\boldsymbol{\theta}_T$, satisfies $|\theta_{Tk}| \geq |\theta_{T(k+1)}|$ for $k = 1, 2, ..., d-1$. We recall that $\xi_k \doteq \Phi^{-1}(1 - kq/(2d))$ where $\Phi^{-1}(.)$ is the quantile of the standard normal distribution and $q \in (0,1)$ is the user-fixed $q$-value. The constraint $b \leq 3\gamma_{\text{WL}}/11$ from ineq. (66) now has to hold with

$$b = b_k \doteq \frac{\xi_k}{\max_j |\pi_{jk}|} \quad . \tag{74}$$

Now, fix

$$a \doteq \min \left\{ \frac{3\gamma_{\text{WL}}}{11}, \frac{\Phi^{-1}(1 - q/(2d))}{\min_k \max_j |\pi_{jk}|} \right\} \quad . \tag{75}$$

Remark that

$$\xi_k \doteq \Phi^{-1}\left(1 - \frac{kq}{2d}\right)$$

$$\geq \Phi^{-1}\left(1 - \frac{q}{2d}\right)$$

$$\geq a \min_{k'} \max_j |\pi_{jk'}| \quad . \tag{76}$$

Suppose $q$ is chosen such that

$$\xi_k \leq \frac{3\gamma_{\text{WL}}}{11} \cdot \max_j |\pi_{jk}| \quad , \forall k \in [d] \quad . \tag{77}$$

This ensures $b_k \leq 3\gamma_{\text{WL}}/11$ ($\forall k \in [d]$) in ineq. (66), while ineq. (76) ensures

$$\Lambda_k \geq b_k \sum_{t:\iota(t)\in\mathcal{F}_k} r_t^2 \tag{78}$$

$$\geq \frac{\xi_k}{\min_{k'} \max_j |\pi_{jk'}|} \cdot \sum_{t:\iota(t)\in\mathcal{F}_k} r_t^2 \tag{79}$$

$$\geq a|\mathcal{F}_k|\gamma_{\text{WL}}^2 \quad . \tag{80}$$

Ineq. (78) holds because of ineqs (73) and (64). Ineq. (80) holds because of the weak learning assumption and ineq. (77). So, we obtain, under the weak learning assumption,

$$\ell_{\mathsf{r}}^{\mathrm{exp}}(\mathcal{S}_r, \boldsymbol{\theta}, \|.\|_\Phi) \;\; \leq \;\; \exp\left(-\frac{aT\gamma_{\mathrm{WL}}^2}{2}\right) \;\; . \tag{81}$$

Ensuring ineq. (77) is done if, after replacing $\xi_k$ by its expression and reorganising, we can ensure

$$q \;\; \geq \;\; 2 \cdot \max_k \frac{q_{N,k}}{q_{D,k}} \;\; , \tag{82}$$

with

$$(0,1) \ni q_{N,k} \;\; \doteq \;\; 1 - \Phi\left(\frac{3\gamma_{\mathrm{WL}}}{11} \cdot \max_j |\pi_{jk}|\right) \;\; , \tag{83}$$

$$(0,1] \ni q_{D,k} \;\; \doteq \;\; \frac{k}{d} \;\; . \tag{84}$$

$$\tag{85}$$

# 3 Supplementary Material on Experiments

## 3.1 Test errors, complete results

To save space, Table 1 below reports only the *lowest* error of all of ADABOOST variants.

## 3.2 Supports for rados (complement to Table 1)

Table 2 in this Supplementary Information provides the supports used to summarize Table 1.

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

Table 1: **Best** result of ADABOOST/regularized-ADABOOST [Schapire and Singer, 1999, Xi et al., 2009], vs $\Omega$-R.ADABOOST (with or without regularization, trained with $n = m$ random rados (above bold horizontal line) or $n = 10000$ rados (below bold horizontal line)), according to the expected true error. Table shows the best result over all $\omega$s, as well as the difference between the worst and best ($\triangle$). Shaded cells display the best result of $\Omega$-R.ADABOOST. For each domain, the sparsest of $\Omega$-R.ADABOOST's method (in average) is indicated with "$\circ$", and the least sparse is indicated with "$\bullet$". When ADABOOST (resp. $\ell_1$-ADABOOST) yields the least sparse (resp. the sparsest) of *all* methods (including $\Omega$-R.ADABOOST), it is shown with "$\blacksquare$" (resp. "$\diamond$"). All domains but Kaggle are UCI [Bache and Lichman, 2013]. The best test error (and std deviation) of ADABOOST and reg.-ADABOOST on Smartphone is zero only up to the second decimal.

| domain | $m$ | $d$ | ADABOOST△ / reg.-ADABOOST err±σ | $\omega = 0$ err±σ | $\Omega = \|\cdot\|_d^2$ err±σ | △ | $\Omega = \|\cdot\|_1$ err±σ | △ | $\Omega = \|\cdot\|_\infty$ err±σ | △ | $\Omega = \|\cdot\|_\Phi$ err±σ | △ |
|---|---|---|---|---|---|---|---|---|---|---|---|---|
| Fertility | 100 | 9 | ◇ 40.00±14.1 | 40.00±14.9 | 41.00±16.6 | 8.00 | ● 41.00±14.5 | 4.00 | ○ 41.00±21.3 | 6.00 | 38.00±14.0 | 7.00 |
| Sonar | 208 | 60 | ■◇ 24.57±9.11 | ● 27.88±4.33 | 25.05±7.56 | 8.14 | 24.05±8.41 | 4.83 | 24.52±8.65 | 10.12 | ○ 25.00±13.4 | 3.83 |
| Spectf | 267 | 44 | ■◇ 45.67±11.0 | ○ 44.96±8.27 | ● 43.43±11.7 | 3.35 | 44.57±12.5 | 1.85 | 43.09±11.0 | 1.85 | 43.79±13.9 | 3.05 |
| Ionosphere | 351 | 33 | ◇ 13.11±6.36 | ● 14.51±7.36 | 13.64±5.99 | 5.43 | 14.24±6.15 | 2.83 | ○ 13.38±4.44 | 3.15 | ○ 14.25±5.04 | 3.41 |
| Breastwisc | 699 | 9 | ■◇ 3.00±1.96 | 3.43±2.25 | 2.57±1.62 | 1.14 | 3.29±2.24 | 0.86 | 2.86±2.13 | 0.86 | ● 3.00±2.18 | 0.29 |
| Transfusion | 748 | 4 | ■◇ 39.17±7.01 | ○ 37.97±7.42 | 37.57±5.60 | 2.40 | ○ 36.50±6.78 | 2.14 | 37.43±8.08 | 1.21 | ● 36.10±8.06 | 3.21 |
| Qsar | 1 055 | 31 | ■◇ 22.09±3.73 | 24.37±4.06 | 22.47±3.84 | 3.41 | ● 23.13±2.74 | 2.75 | 23.23±3.64 | 2.64 | ● 23.80±3.79 | 2.55 |
| Hill-nonoise | 1 212 | 100 | ◇ 47.52±5.14 | 45.63±6.68 | 45.71±6.61 | 0.33 | 45.46±6.88 | 0.66 | ○ 45.87±6.69 | 0.49 | ● 45.62±7.26 | 0.42 |
| Hill-noise | 1 212 | 100 | ◇ 47.61±3.48 | ○ 45.05±2.98 | 44.80±2.86 | 0.99 | 44.97±3.37 | 0.66 | 44.88±2.82 | 0.66 | ● 44.64±3.26 | 0.74 |
| Winered | 1 599 | 11 | ■◇ 26.33±2.75 | ○ 28.02±3.32 | 27.83±3.95 | 1.19 | ● 27.45±4.17 | 1.00 | ○ 27.58±3.76 | 1.12 | ● 27.45±3.34 | 1.25 |
| Abalone | 4 177 | 10 | ■◇ 22.98±2.70 | ● 26.57±2.31 | 24.18±2.51 | 0.00 | 24.13±2.48 | 0.14 | ○ 24.18±2.51 | 0.00 | ● 24.11±2.59 | 0.07 |
| Statlog | 4 435 | 36 | ■ 4.49±0.61 | 22.41±2.20 | ○ 4.71±0.82 | 0.25 | ○ 20.43±1.89 | 0.23 | 4.69±0.72 | 0.45 | 20.00±1.80 | 0.18 |
| Winewhite | 4 898 | 11 | ■◇ 30.73±2.20 | ● 32.63±2.52 | 31.85±1.66 | 1.18 | 32.16±1.73 | 1.31 | 32.16±2.02 | 0.90 | 31.97±2.26 | 1.12 |
| Smartphone | 7 352 | 561 | ■ 0.00±0.00 | ○ 0.67±0.25 | 0.19±0.22 | 0.00 | ○ 0.44±0.29 | 0.03 | 0.20±0.24 | 0.01 | 0.19±0.22 | 0.04 |
| Firmteacher | 10 800 | 16 | ■◇ 44.44±1.34 | ● 40.58±4.87 | 40.89±3.95 | 2.35 | 39.81±4.37 | 2.89 | ● 38.91±4.51 | 3.56 | ● 38.01±6.15 | 5.02 |
| Eeg | 14 980 | 14 | ◇ 45.38±2.04 | 44.09±2.32 | ○ 44.01±1.48 | 0.40 | 43.89±2.19 | 0.89 | 44.07±2.02 | 0.81 | ● 43.87±1.40 | 0.95 |
| Magic | 19 020 | 10 | ■◇ 21.07±1.09 | ○ 37.51±0.46 | 22.11±1.32 | 0.28 | ● 26.41±1.08 | 0.00 | 23.00±1.71 | 0.66 | ○ 26.41±1.08 | 0.00 |
| Hardware | 28 179 | 96 | 16.77±0.73 | ○ 9.41±0.71 | 6.43±0.74 | 0.18 | ○ 11.72±1.24 | 0.41 | ● 6.50±0.67 | 0.10 | 6.42±0.69 | 0.13 |
| Marketing | 45 211 | 27 | ■ 30.68±1.01 | ○ 27.70±0.69 | 27.33±0.73 | 0.33 | ○ 28.02±0.47 | 0.00 | ● 27.19±0.87 | 0.51 | ○ 28.02±0.47 | 0.00 |
| Kaggle | 120 269 | 10 | ■ 47.80±0.47 | ● 39.22±8.47 | ○ 16.90±0.51 | 0.00 | ○ 16.90±0.51 | 0.00 | ● 16.89±0.50 | 0.01 | 16.90±0.51 | 0.00 |

| domain | $m$ | $d$ | AdaBoost supp.$\pm\sigma$ | reg.-AdaBoost supp.$\pm\sigma$ | $\Omega$-R.AdaBoost $\omega=0$ supp.$\pm\sigma$ | $\Omega=\|\cdot\|^2_d$ supp.$\pm\sigma$ | $\Omega=\|\cdot\|_1$ supp.$\pm\sigma$ | $\Omega=\|\cdot\|_\infty$ supp.$\pm\sigma$ | $\Omega=\|\cdot\|_\Phi$ supp.$\pm\sigma$ |
|---|---|---|---|---|---|---|---|---|---|
| Fertility | 100 | 9 | 36.67±36.3 | 14.44±5.36 | 37.78±31.1 | 36.67±34.8 | ●42.22±31.0 | ○24.44±22.1 | 32.22±17.7 |
| Sonar | 208 | 60 | 57.83±3.69 | 1.83±0.52 | ●14.17±3.62 | 14.00±3.16 | 13.67±3.99 | 12.83±4.45 | ○12.67±4.17 |
| Spectf | 267 | 44 | 56.14±6.34 | 2.27±0.00 | ○13.41±5.71 | ●15.91±6.94 | 13.64±6.60 | 15.00±6.53 | 13.64±6.15 |
| Ionosphere | 351 | 33 | 76.97±8.23 | 3.64±1.27 | ●13.64±3.85 | 13.03±3.51 | 11.82±3.63 | ○11.21±4.75 | 11.21±4.53 |
| Breastwisc | 699 | 9 | 90.00±3.51 | 11.11±0.00 | 51.11±12.0 | 84.44±7.77 | ○48.89±9.37 | 84.44±10.7 | ●86.67±4.68 |
| Transfusion | 748 | 4 | 77.50±14.2 | 25.00±0.00 | ○67.50±20.6 | 70.00±23.0 | ○67.50±16.9 | ○67.50±23.7 | ●72.50±14.2 |
| Qsar | 1 055 | 31 | 43.66±2.92 | 4.88±0.00 | 17.80±3.82 | 17.07±5.27 | ●18.29±4.34 | ●18.29±4.34 | ○16.34±3.64 |
| Hill-nonoise | 1 212 | 100 | 3.30±0.67 | 1.10±0.32 | 3.40±0.52 | 3.30±0.48 | 3.40±1.07 | ○3.20±0.92 | ●12.40±3.98 |
| Hill-noise | 1 212 | 100 | 4.00±0.32 | 1.10±0.32 | ○5.60±0.70 | 6.70±1.34 | 6.00±1.63 | 5.70±0.82 | ●6.80±1.55 |
| Winered | 1 599 | 11 | 79.09±6.14 | 9.09±0.00 | ○25.45±5.75 | ●27.27±6.06 | ●27.27±7.42 | ○25.45±7.17 | 27.27±7.42 |
| Abalone | 4 177 | 10 | 64.00±6.99 | 19.00±3.16 | ●30.00±6.67 | ○10.00±0.00 | 12.00±6.32 | ○10.00±0.00 | 11.00±3.16 |
| Statlog | 4 435 | 36 | 48.89±22.2 | 3.06±0.88 | 20.07±1.90 | ●39.44±19.3 | ○2.78±0.00 | 33.33±0.00 | 25.00±0.00 |
| Winewhite | 4 898 | 11 | 66.36±9.63 | 9.09±0.00 | ●28.18±2.87 | ●28.18±2.87 | 20.91±4.39 | 27.27±0.00 | ○18.18±0.00 |
| Smartphone | 7 352 | 561 | 5.53±0.24 | 0.36±0.00 | ○0.18±0.00 | 71.21±20.1 | ○0.18±0.00 | ●74.72±19.7 | 24.69±9.87 |
| Firmteacher | 10 800 | 16 | 48.12±30.8 | 10.00±3.22 | 24.38±7.48 | ●25.62±9.52 | 21.25±4.37 | ○20.62±4.22 | 20.62±9.34 |
| Eeg | 14 980 | 14 | 14.29±3.37 | 8.57±3.01 | 39.29±13.2 | ○38.57±9.04 | ●39.29±14.0 | ○38.57±13.1 | ●39.29±10.8 |
| Magic | 19 020 | 10 | 45.00±7.07 | 10.00±0.00 | ○10.00±0.00 | ●51.00±3.16 | ○10.00±0.00 | 49.00±7.38 | ●10.00±0.00 |
| Hardware | 28 179 | 96 | 11.98±7.56 | 2.19±0.33 | ○1.04±0.00 | 20.94±3.12 | ●1.04±0.00 | 22.08±1.89 | 21.25±1.49 |
| Marketing | 45 211 | 27 | 65.19±3.58 | 7.40±0.00 | ●7.41±0.00 | 12.96±3.60 | ○3.70±0.00 | ●13.33±4.35 | ○3.70±0.00 |
| Kaggle | 120 269 | 10 | 28.18±5.16 | 18.18±0.00 | ●17.27±2.87 | ○9.09±0.00 | 15.45±4.39 | 10.00±2.87 | 14.55±4.69 |

Table 2: Supports of AdaBoost and $\ell_1$-AdaBoost vs $\Omega$-R.AdaBoost for the results displayed in Table 1 of the main file (supp.%$(\boldsymbol{\theta})$ $\doteq 100 \cdot \|\boldsymbol{\theta}\|_0 / d$). For each domain, the sparsest of $\Omega$-R.AdaBoost's method (in average) is indicated with "○", and the least sparse is indicated with "●".