[Reviews · NeurIPS 2016]

Reviewer 1

Summary

The authors extend and generalize previous work on learning with Rademacher observations. In particular - The authors provide a unified theoretical framework that shows that a number of example losses have equivalent Rademacher observation losses. This extends previous work which presented equivalents for the square and logistic losses. - The authors prove that a regularizer in the example loss can be transferred to the Rado loss. - Based on the above, the authors propose a novel boosting algorithm that minimizes a regularized Rademacher observation loss, showing that learning is computationally efficient. - The authors also provide empirical evidence to the potential practical importance of the proposed boosting algorithm.

Qualitative Assessment

The presented work is of very high quality and definitely has a place at NIPS. As indicated by my scoring my only qualm would be with respect to its potential impact or usefulness. Looking at the, excellent, work of Nock or Patrini, it does not seem to have influenced the work of many other researchers in ensemble learning or other fields. As such I am not entirely positive that the presented work would be of interest to a large number of attendees; the practical impact of the work is also not clear, though the empirical results (on UCI datasets) would seem to indicate some practical importance. If the authors could point out some citations showing the impact or general usefulness of the work (or similar work,e.g. Nock's work), it would be helpful.

Confidence in this Review

2-Confident (read it all; understood it all reasonably well)


Reviewer 2

Summary

This paper showed that the equivalences between two example and rado losses can be unified and generalized. Moreover, this equivalence can be extended to regularized losses. A sufficient condition is introduced to guide when and how regularizing the loss over examples is equivalent to regularizing the rados in the equivalent rado loss. A formal boosting algorithm for the regularized exponential rado-loss is proposed to boost with any of the ridge, lasso, SLOPE, L1, or elastic net regularizer using the same master routine. Experiments demonstrated that regularization improves rado-based learning.

Qualitative Assessment

Pro: - It was reasonably proved that the regularized exponential rado-loss is the equivalent of the regularized logistic loss over examples - A formal boosting algorithm is proposed for the regularized exponential rado-loss. Experimental results demonstrated its advantages over the versions without regularization. Con: - While experimental results have clearly shown that regularization versions could achieve better results over none-regularization version, the current results are not enough to confirm the conclusion: \omega-R-Adaboost is better than Adaboost. More experiments are preferred here. - In bigger domains, \omega-R-Adaboost is much better than Adaboost. The former processed data randomly selected from the whole data samples, whole the latter processed all data samples. It is interesting to see how about the latter only processed data randomly selected from the whole data samples. In this case, the former is still much better? - typo: line 8: “bis repetita placent” ?? line 32: comparatively significantly inferior??? ……

Confidence in this Review

2-Confident (read it all; understood it all reasonably well)


Reviewer 3

Summary

The paper makes contributions involving Rademacher observations (previously dubbed "rados"), and their relationship to regularization. It shows that the compression of the dataset's information into the set of rados has a regularizing effect, defines rado-losses which are of special importance in the theory, and demonstrates they hold practical value.

Qualitative Assessment

The technical contribution of this paper is clear - I recommend acceptance because of the potential value. The theorems apply only to the four losses mentioned, and I was not able to go through the proofs in full detail so I have some doubts about how they would generalize to efficient algorithms otherwise; however, I had more significant doubts after reading the original Nock et al. 2015 paper, and clearly the theory has grown since then. It is indeed interesting that the new boosting algorithm can be implemented as efficiently as the standard ones, but I view this as a side consequence of the formulation and have therefore focused on the remainder in my evaluation. There are several places in the paper with typos (e.g. line 101, 217), and the material can get dense for a general machine learning audience, though it is mathematically clean and precise. E.g., the discussion of the affinely generated functions in Sec. 2 would be much clearer if re-ordered, so that the motivation for the definitions and choices is clear (e.g. rados represent powersets, so the powerset indexing would clear up significantly). For similar reasons I suggest defining (1) and (2) as a different script letter instead of L, due to Eqs. (11,12) and the possible confusion with losses. Following from the paper are several interesting and open questions with possible connections to other areas of machine learning. I am interested in the connection to the two-player game formulation, because it suggests different parametrizations of the problem by choosing the z's differently for other learning settings. The z's are currently chosen in a similar way to the boosting game, this conflation of information happens due to linear constraints on the z_i's (weak learning assumption), so they are chosen in this setting with benefits. But the results in Sec. 2 are general and there are other similar game-theoretic formulations in areas like (e.g. online) learning. Are there other ways of condensing the data, besides rados, that are similarly beneficial for those problems?

Confidence in this Review

2-Confident (read it all; understood it all reasonably well)


Reviewer 4

Summary

The paper is about learning in the framework of so-called rado (RADemacher Observation) losses. The idea of this approach to learning is to replace the usual learning over examples to learning over their averages over random subsets. For such a replacement to be possible for some specific loss function ('example loss'), one has to point out the counterpart rado loss such that both losses, as functions of a classifier, are linked by a monotone function, and hence one can minimize the rado loss instead of the example loss. As explained in [Nock et al. 2015], rado learning can be beneficial in the context of differential privacy. Previously in the literature, equivalent losses were known only for the logistic and the quadratic loss. Authors provide a unified framework to prove such equivalence, and use it to establish the counterpart rado losses for a number of other loss functions. They also formulate a sufficient condition under which the equivalence retains when the example loss is regularized. As a final contribution, they provide a boosting algorithm for the exponential rado loss regularized by numerous common regularizers.

Qualitative Assessment

Authors use their unified framework to find the equivalent rado losses to a number of common loss functions. As such, the results will probably be useful within the differential privacy community. I had some troubles understanding the first part of the paper (Sec. 2) because of how the exposition was organised, namely, that the learning setting is introduced only after the zero-sum game notation.

Confidence in this Review

1-Less confident (might not have understood significant parts)